# DNA methylation insulates genic regions from CTCF loops near nuclear speckles

**Shelby A Roseman**[1,2,3†], **Allison P Siegenfeld**[1,2,4†], **Ceejay Lee**[1,2], **Nicholas Z Lue**[1,2,5], **Amanda L Waterbury**[1,2], **Brian B Liau**[1,2]*

[1]Harvard University Department of Chemistry and Chemical Biology, Cambridge, United States; [2]Broad Institute of MIT and Harvard, Cambridge, United States; [3]Dana-Farber Cancer Institute, Boston, United States; [4]Harvard Medical School, Boston, United States; [5]University of California, Berkeley, Berkeley, United States

## eLife Assessment

This **valuable** study tested the impact of DNA methylation on CTCF binding in two cancer cell lines. Increased CTCF binding sites are enriched in gene bodies, and associate with nuclear speckles, indicating a role in increased transcription. In the revised work, the inferred association with nuclear speckles has been supported with more **solid** data. These results will be of interest to the epigenetics field.

*For correspondence:
liau@chemistry.harvard.edu

†These authors contributed equally to this work

## Abstract

The insulator protein CTCF is essential for mediating chromatin loops and regulating gene expression. While it is established that DNA methylation hinders CTCF binding, the impacts of this methylation-sensitive CTCF binding on chromatin architecture and transcription are poorly defined. Here, we used a selective DNMT1 inhibitor (DNMT1i) to investigate the characteristics and functions of 'DNMT1i-specific' CTCF peaks resulting from global DNA demethylation. We found that DNMT1i-specific peaks preferentially form chromatin loops on gene bodies and interact with highly looping partner peaks located in regions of active chromatin. Notably, both DNMT1i-specific CTCF peaks and their highly looping partners are enriched near nuclear speckles – condensate bodies implicated in transcription and splicing. Utilizing targeted protein degradation, we specifically depleted CTCF and nuclear speckles to elucidate their functional interplay. By degrading CTCF upon DNMT1 inhibition, we revealed that CTCF is important for DNMT1i-dependent interactions between chromatin and speckle proteins. Moreover, we found that CTCF promotes the activation of genes near speckles upon DNMT1 inhibition. Conversely, acute depletion of nuclear speckles revealed that they influence RNA abundance but do not maintain CTCF binding or looping. Collectively, our study suggests a model wherein DNA methylation prevents spurious CTCF occupancy and interactions with regulatory elements near nuclear speckles, yet CTCF looping is robust toward the loss of speckles.

## Introduction

The mammalian genome is organized at multiple length scales into compartments, topologically associating domains (TADs), and chromatin loops (*Rowley and Corces, 2018*; *Rao et al., 2014*; *Dixon et al., 2012*; *Misteli, 2020*). This three-dimensional chromatin folding contributes to essential cellular processes such as transcription and DNA replication and is actively maintained by protein cofactors (*Zheng and Xie, 2019*; *Rao et al., 2017*; *Nora et al., 2017*; *van Steensel and Belmont, 2017*). The architectural protein CTCF establishes the anchors of many DNA loops and TADs likely by blocking loop extrusion by cohesin (*Nora et al., 2017*; *Wendt et al., 2008*; *Li et al., 2020*; *Fudenberg et al.,*

*2016*; *Kim et al., 2019*; *Davidson et al., 2019*; *Xiang and Corces, 2021*; *Hansen et al., 2018*; *Sanborn et al., 2015*). Consequently, perturbation of CTCF binding sites at domain boundaries can lead to aberrant cross-domain enhancer-promoter contacts, which have been linked to gene misexpression in cancer and developmental disorders (*Dubois et al., 2022*; *Spielmann et al., 2018*; *Lupiáñez et al., 2015*; *Flavahan et al., 2016*; *Flavahan et al., 2019*). Furthermore, CTCF binding and looping on or near genes can directly enforce enhancer-promoter loops and facilitate differential splicing and polyadenylation (*Rinzema et al., 2022*; *Kubo et al., 2021*; *Shukla et al., 2011*; *Ruiz-Velasco et al., 2017*; *Nanavaty et al., 2020*; *Alharbi et al., 2021*; *López Soto and Lipscombe, 2020*; *van Steensel and Furlong, 2019*). Although these examples demonstrate that CTCF can modulate transcription in specific contexts, acute depletion of CTCF only leads to moderate transcriptional defects, highlighting the complex and nuanced roles of CTCF in regulating gene expression (*Rao et al., 2017*; *Nora et al., 2017*; *Schoenfelder and Fraser, 2019*; *Vermunt et al., 2019*).

CTCF binding is sensitive to DNA methylation (*Monteagudo-Sánchez et al., 2024a*), and consequently, global DNA demethylation causes the emergence of hundreds to thousands of previously masked CTCF peaks (*Bell and Felsenfeld, 2000*; *Hark et al., 2000*; *Hashimoto et al., 2017*; *Maurano et al., 2015*; *Wang et al., 2012*). However, these methylation-sensitive sites comprise only a small subset of all potential methylated CTCF sites, suggesting that DNA methylation serves as a specialized mechanism to regulate specific CTCF binding sites (*Maurano et al., 2015*). Moreover, recent work has revealed that DNA methylation regulates a subset of CTCF loops and CTCF-dependent genes during mESC differentiation (*Monteagudo-Sánchez et al., 2024b*). While methylation-sensitive CTCF peaks have been linked to gene expression and mRNA processing in some instances (*Flavahan et al., 2016*; *Flavahan et al., 2019*; *Shukla et al., 2011*; *Ruiz-Velasco et al., 2017*; *Nanavaty et al., 2020*; *Monteagudo-Sánchez et al., 2024b*; *Schuijers et al., 2018*), the functions of most methylation-sensitive CTCF peaks remain unknown. Moreover, the broader impacts of emergent CTCF binding upon global DNA demethylation on chromatin looping are just starting to be revealed (*Monteagudo-Sánchez et al., 2024b*).

In this study, we employed a selective inhibitor of DNMT1, the maintenance DNA methyltransferase, to induce genome-wide DNA demethylation and investigated the chromatin looping patterns of the resulting methylation-sensitive DNMT1i-specific CTCF peaks. We found that DNMT1i-specific CTCF peaks within genes form loops to highly looping partner peaks located in active chromatin regions. Remarkably, DNMT1i-specific CTCF peaks and their highly looping partners are located close to nuclear speckles – condensate bodies of protein and RNA that have been implicated in transcription and splicing (*Chen et al., 2018*; *Chen and Belmont, 2019*; *Zhang et al., 2021*; *Wang et al., 2021*; *Spector and Lamond, 2011*; *Alexander et al., 2021*; *Ilık and Aktaş, 2022*). Subnuclear localization relative to nuclear landmarks such as nuclear speckles is linked to genome structure and function, so we explored this relationship further by developing acute degron systems for both CTCF and speckles. Through these experiments, we clarify the directionality of the relationship between CTCF and speckles and glean novel insights into how methylation-sensitive CTCF contributes to chromatin structure and gene expression.

## Results
### DNMT1 inhibition activates CTCF peaks and loops on gene bodies

Loss of DNA methylation leads to increased CTCF occupancy at hundreds to thousands of CTCF sites depending on the cell type (*Maurano et al., 2015*), but a global understanding of their looping patterns and functions is lacking. Recently, selective, noncovalent DNMT1 inhibitors (e.g. GSK3484862) have become available, enabling dynamic control of DNA methylation with fewer undesirable effects such as DNA damage and cytotoxicity (*Azevedo Portilho et al., 2021*; *Pappalardi et al., 2021*). Consequently, we employed these new DNMT1 inhibitors to probe the functions and looping patterns of methylation-sensitive CTCF sites.

We treated K562 and HCT116 cells with GSK3484862 (hereafter referred to as DNMT1i, 10 µM) for 3 days and performed CTCF ChIP-seq and HiChIP to investigate CTCF-associated changes in chromatin looping (*Figure 1A*). Under these conditions, DNMT1i treatment led to minimal growth defects in both cell types (*Figure 1—figure supplement 1A*). Consistent with prior studies (*Maurano et al., 2015*), DNMT1i treatment increased CTCF binding at thousands of sites, which we term

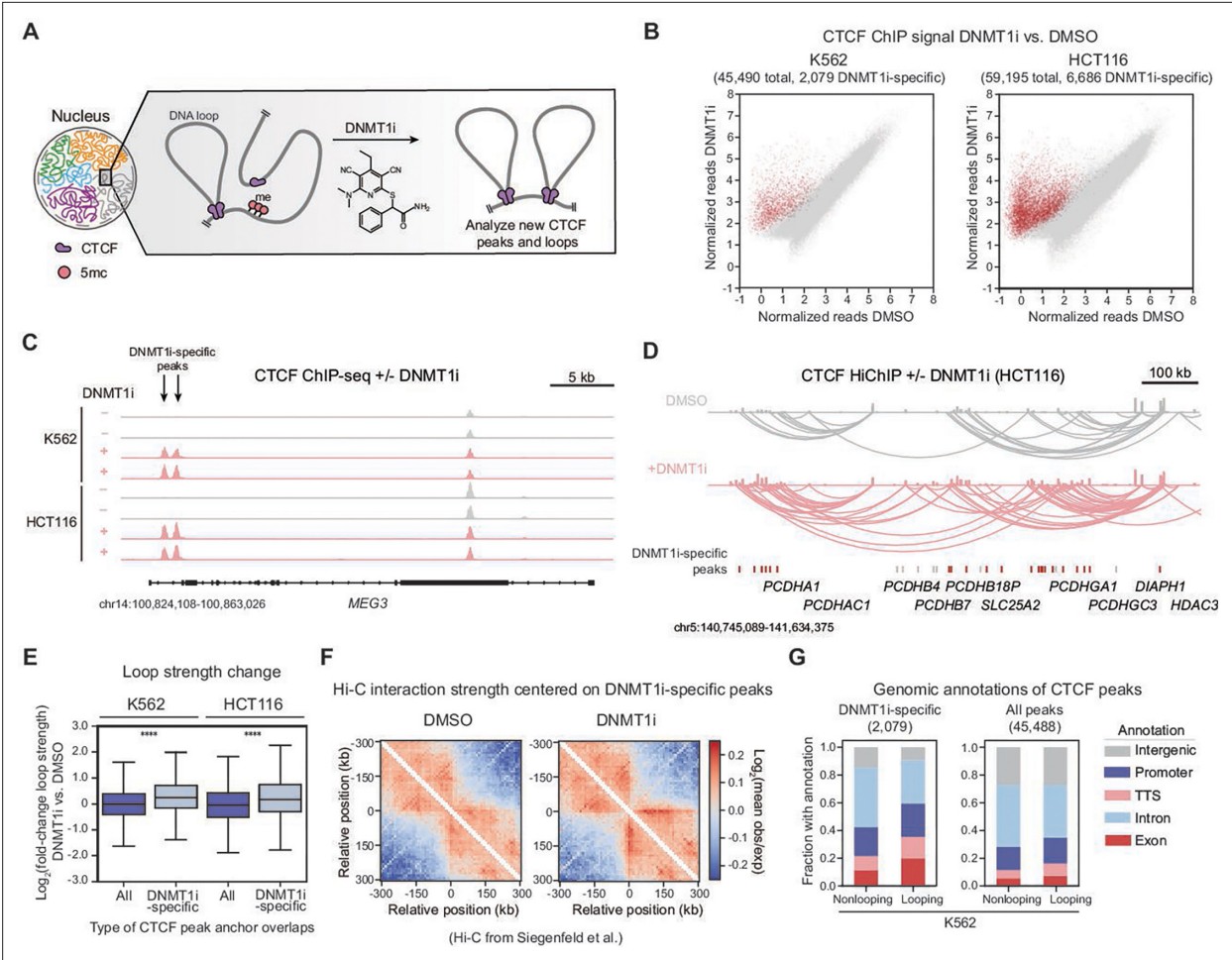

**Figure 1.** DNMT1 inhibition activates CTCF peaks and loops on gene bodies. (**A**) Schematic demonstrating the experimental workflow. K562 and HCT116 cells were treated with DNMT1i for 3 days followed by CTCF ChIP-seq and CTCF HiChIP. (**B**) Replicate-averaged input-normalized reads within merged peaks for CTCF ChIP-seq in DNMT1i (y-axis) vs. DMSO (x-axis) for K562 cells (left) and HCT116 cells (right). DNMT1i-specific peaks are highlighted in red. (**C**) Example Integrated Genome Browser (IGV) CTCF ChIP-seq tracks ± DNMT1i at the MEG3 locus for K562 (top) and HCT116 (bottom). DNMT1i-specific peaks indicated with arrows. (**D**) Example IGV CTCF ChIP-seq and HiChIP tracks ± DNMT1i at the Protocadherin Alpha (*PCDHA*) locus (HCT116). DNMT1i-specific peaks are indicated below. Red indicates a looping DNMT1i-specific peak, whereas gray indicates a nonlooping DNMT1i-specific peak. (**E**) Boxplot showing the log₂(fold-change) in loop strength DNMT1i vs. DMSO (y-axis) for loops with or without at least one anchor overlapping a DNMT1i-specific peak (x-axis) for K562 (left) and HCT116 (right). (**F**) Heatmaps depicting log₂(mean observed/expected) Hi-C interaction frequency centered on DNMT1i-specific peaks for DMSO (left) and DNMT1i (right). K562 Hi-C data from *Siegenfeld et al., 2022*. (**G**) Proportion of looping and nonlooping peaks with each annotation (y-axis) for both DNMT1i-specific and all CTCF peaks in K562 in DNMT1i. In (**E**), the interquartile range (IQR) is depicted by the box with the median represented by the center line. Outliers are excluded. P-Values were calculated by a Mann-Whitney test and are annotated as follows: ns: not significant; *: 0.01<p≤0.05; **: 0.001<p≤0.01; ***: 0.0001<p≤0.001; ****: p≤0.0001.

The online version of this article includes the following figure supplement(s) for figure 1:

**Figure supplement 1.** DNMT1 inhibition activates CTCF peaks and loops on gene bodies.

'DNMT1i-specific peaks' (2079 up- and 2 downregulated peaks in K562, 6686 up- and 328 downregulated peaks in HCT116). These peaks are known to overlap with CTCF peaks in other cell types (*Maurano et al., 2015*; *Spracklin et al., 2023*; *Figure 1B and C*). As expected, DNMT1i-specific CTCF sites were 80–90% methylated in wild-type cells (*Figure 1—figure supplement 1B*), and half of the CTCF binding motifs enriched within these peak regions were methylated at two key CpG sites implicated in CTCF binding (*Figure 1—figure supplement 1C*; *Wang et al., 2012*; *Spracklin et al., 2023*). Analysis of genome-wide bisulfite information obtained from LIMe-Hi-C in K562 treated with DMSO and DNMT1i (*Figure 1—figure supplement 1D*) revealed that these DNMT1i-specific CTCF sites at baseline are more methylated than their constitutive counterparts (*Figure 1—figure supplement*

*1E*). Strikingly, upon treatment with the DNMT1i treatment for 3 days, methylation levels at DNMT1i-specific CTCF sites are reduced twofold (*Figure 1—figure supplement 1E*).

We next evaluated whether these DNMT1i-specific CTCF peaks form chromatin loops. CTCF HiChIP revealed that a large subset of DNMT1i-specific peaks (45% K562, 30% HCT116) resides within at least one loop anchor and these are henceforth called 'looping' peaks (see Methods) (*Lareau and Aryee, 2018a*; *Lareau and Aryee, 2018b*). An example of DNMT1i-specific looping CTCF peaks is shown at the *PCDHA* locus (*Figure 1D*). On average, loops that are anchored by DNMT1i-specific peaks increase moderately in strength upon DNMT1 inhibition (average log$_2$fold-change~0.3) (*Figure 1E*). We next sought to evaluate these findings using our published LIMe-Hi-C dataset following DNMT1 inhibition (*Siegenfeld et al., 2022*). Specifically, we analyzed Hi-C interaction strength using standard aggregate pileup analysis over all DNMT1i-specific CTCF sites identified in our ChIP-seq data. Consistent with our Hi-ChIP data, this analysis revealed an increase in Hi-C interaction signal at DNMT1i-specific peaks following 3 days of DNMT1 inhibition (*Figure 1F*).

We next characterized the genomic annotations of looping and nonlooping DNMT1i-specific CTCF peaks. Consistent with previous findings using azacitidine (*Maurano et al., 2015*), loss of DNA methylation through DNMT1i treatment led to the preferential reactivation of CTCF peaks on non-intron gene bodies, with half of K562 DNMT1i-specific peaks emerging in exons, promoters, or transcription termination sites (*Figure 1G*, *Figure 1—figure supplement 1F and G*). Notably, this enrichment on gene bodies was particularly pronounced with looping DNMT1i-specific CTCF peaks (*Figure 1G*, *Figure 1—figure supplement 1F, G, and H*), highlighting an underappreciated propensity for CTCF peaks on gene bodies to engage in chromatin loops. Altogether, selective DNMT1 inhibition leads to the emergence of thousands of DNMT1i-specific CTCF peaks on gene bodies that engage in new chromatin interactions.

## DNMT1i-specific CTCF peaks interact with highly looping partners near nuclear speckles

We next scrutinized the CTCF partners that loop to DNMT1i-specific CTCF peaks. Notably, the CTCF partners of DNMT1i-specific peaks tend to engage in a moderately greater total number of different loops compared to all CTCF peaks (i.e. have more looping partners, *Figure 2A*), mostly for K562 cells (average increase ~1.3-fold). This trend is in part driven by the tendency of non-intronic gene body CTCF peaks to more frequently participate in loops with highly looping CTCF partners in comparison to intronic or intergenic CTCF peaks (*Figure 2—figure supplement 1A*). To characterize these highly looping partners, we ranked all CTCF peaks by the number of different partners they interact with and called those with the most partners 'highly looping', analogously to how superenhancers or supersilencers were previously characterized (*Figure 2B and C*, *Figure 2—figure supplement 1B*; *Cai et al., 2021*; *Hnisz et al., 2013*). HiChIP loops between DNMT1i-specific peaks and highly looping peaks also increased in strength upon DNMT1 inhibition (average log$_2$fold-change~0.3), consistent with increased interactions between these sites (*Figure 2—figure supplement 1C*). The propensity for a given CTCF site to engage in many contacts may reflect architectural stripes, a chromatin structural feature identified from Hi-C data that is postulated to result from persistent loop extrusion at a single region (*Vian et al., 2018*; *Yoon et al., 2022*; *Barrington et al., 2019*). Stripe anchors are enriched for chromatin accessibility and histone modifications associated with active enhancers (*Vian et al., 2018*). Similarly, regions near highly looping CTCF peaks are accessible and modestly enriched for active chromatin marks such as H3K9ac and H3K27ac (*Figure 2D*). Furthermore, based on previously published Hi-C data (*Siegenfeld et al., 2022*), highly looping CTCF sites have an approximately twofold stronger propensity to engage in chromatin interactions compared to ordinary CTCF sites (*Figure 2E*). In addition, stripe anchors identified from these Hi-C data also reside closer to highly looping CTCF peaks than all peaks, consistent with the similarities between these features (*Figure 2—figure supplement 1D*). Thus, these results show that upon DNA demethylation, DNMT1i-specific CTCF sites form loops to highly looping partners associated with active regulatory marks and stripe anchors.

To understand the features that distinguish DNMT1i-specific CTCF peaks and their highly looping partners from other CTCF peaks, we next considered their subnuclear localization. To do so, we investigated their distribution across genomic subcompartments, which exhibit variable transcriptional activity and association with nuclear landmarks, such as the nuclear lamina or nuclear speckles. In

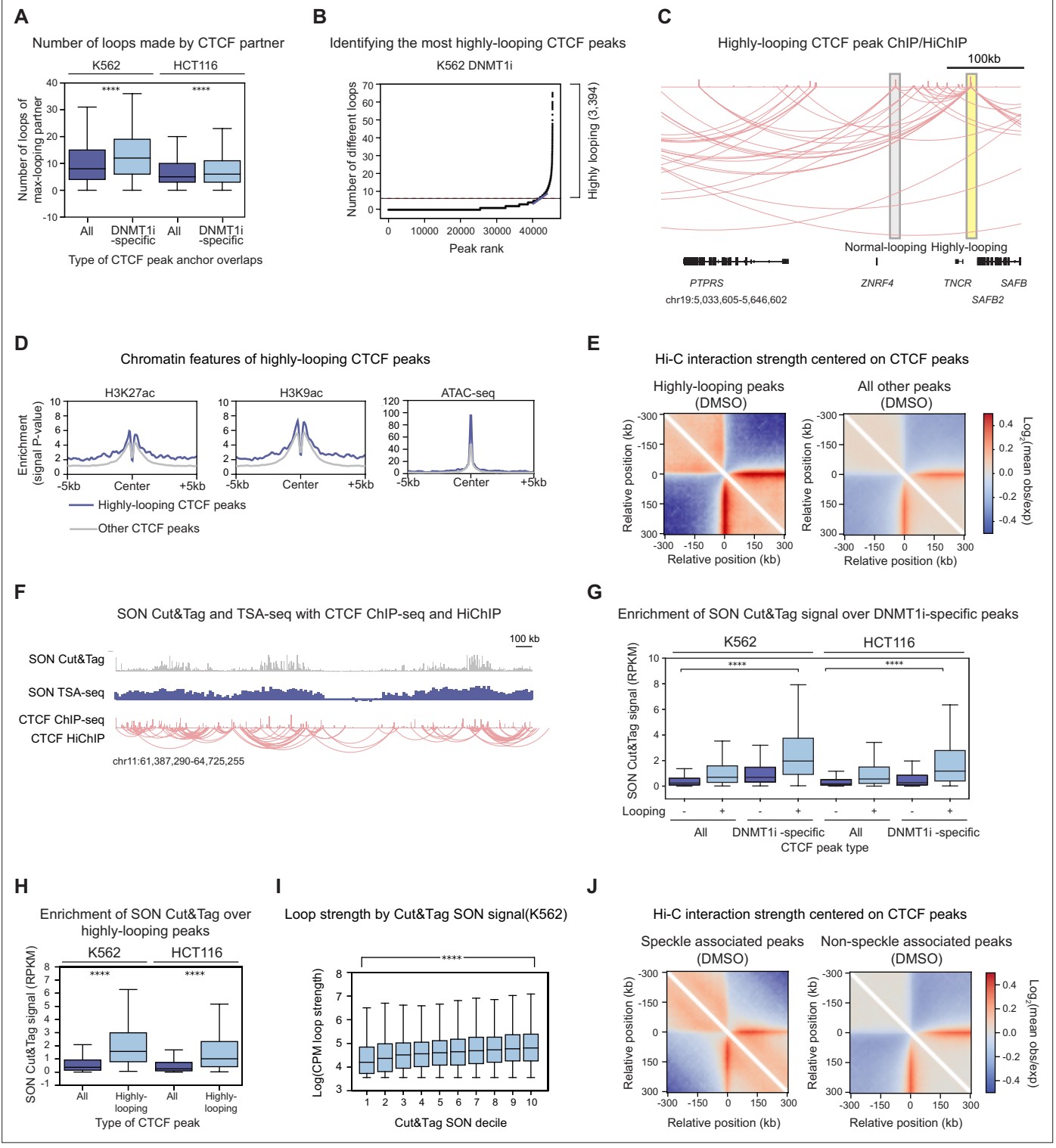

**Figure 2.** DNMT1i-specific CTCF peaks interact with highly looping partners near nuclear speckles. (**A**) Number of different loops (loops to different partners) the maximally looping partner peak makes (y-axis) for all vs. DNMT1i-specific CTCF peaks (x-axis) in DNMT1i for K562 (left) and HCT116 (right). (**B**) Number of different loops (number of partner peaks) (y-axis) assigned to each peak by rank (x-axis) in DNMT1i for K562. Highly looping peaks reside above the red dashed line. (**C**) Integrated Genome Browser (IGV) representation of highly looping (yellow) and normal-looping CTCF peaks (gray) in K562 DNMT1i-treated cells. CTCF ChIP-seq and HiChIP are shown. (**D**) Aggregate profile (signal p-value, y-axis) of genomic features over highly looping peaks (blue) and all other CTCF peaks (gray). CTCF peaks were called in K562 DNMT1i treatment, and histone modifications/ATAC-seq data are from public datasets in K562 (see Methods). (**E**) Heatmap depicting log₂(mean observed/expected) Hi-C interaction frequency centered on highly looping

*Figure 2 continued on next page*

*Figure 2 continued*

(left) and all other (right) peaks in the DMSO treatment condition in K562 cells. K562 Hi-C data from *Siegenfeld et al., 2022*. (F) Example IGV tracks depicting SON Cut&Tag signal (top), SON TSA-seq normalized counts (middle), CTCF ChIP-seq (bottom), and CTCF HiChIP for all CTCF peaks for a region on chromosome 11 for K562 DMSO. (G) Boxplot showing replicate-averaged DNMT1i SON Cut&Tag signal (RPKM, 20 kb bins, y-axis) in the respective cell type over DNMT1i-specific peaks vs. all other CTCF peaks called in DNMT1i for K562 and HCT116 cells broken down by whether the CTCF peak is in a loop anchor (x-axis). (H) Same as G, but for highly looping peaks vs. all other CTCF peaks. (I) Boxplot showing average $\log_2$(counts per million [CPM] loop strength) (y-axis) for CTCF HiChIP loops relative to SON Cut&Tag signal decile (x-axis). logCPM defined by Diffloop across all conditions (DMSO and DNMT1i). Loops are segregated into equally sized deciles by Cut&Tag signal in DMSO (RPKM, 20 kb bins). Pearson R=0.2. (J) Heatmap depicting $\log_2$(mean observed/expected) Hi-C interaction frequency centered on CTCF peaks at speckles (denoted by high SON signal, left) and not at speckles (right) in DMSO. K562 Hi-C data from *Siegenfeld et al., 2022*. In (A, G, H, and I), the interquartile range (IQR) is depicted by the box with the median represented by the center line. Outliers are excluded. P-Values were calculated by a Mann-Whitney test and are annotated as follows: ns: not significant; *: $0.01 < p \leq 0.05$; **: $0.001 < p \leq 0.01$; ***: $0.0001 < p \leq 0.001$; ****: $p \leq 0.0001$.

The online version of this article includes the following figure supplement(s) for figure 2:

**Figure supplement 1.** DNMT1i-specific CTCF peaks interact with highly looping partners near nuclear speckles.

particular, we found an enrichment of DNMT1i-specific CTCF peaks and highly looping CTCF peaks in the A1 subcompartment of the genome, which is differentiated from the A2 subcompartment largely due to its localization near nuclear speckles (*Figure 2—figure supplement 1E*; *Xiong and Ma, 2019*). Studies have shown that nuclear speckles are hubs of DNA-DNA contacts (*Spector and Lamond, 2011*; *Ilık and Aktaş, 2022*; *Brown et al., 2008*; *Ilik et al., 2020*; *Faber et al., 2022*; *Quinodoz et al., 2018*; *Kim et al., 2020*). Additionally, CTCF can form stress-responsive clusters near nuclear speckles in some cell types, and genomic subcompartments associated with speckles have been correlated with increased CTCF binding and looping (*Chen et al., 2018*; *Wang et al., 2021*; *Jabbari et al., 2019*; *Lehman et al., 2021*). However, a direct connection between nuclear speckles and CTCF binding and looping has not been extensively explored.

To determine if DNMT1i-specific peaks and highly looping CTCF peaks are preferentially located near nuclear speckles, we performed Cut&Tag for SON, a core speckle protein, in K562 and HCT116 cells with and without DNMT1 inhibition (*Figure 2F*; *Chen et al., 2018*; *Zhang et al., 2021*; *Ilik et al., 2020*; *Kaya-Okur et al., 2019*). Consistent with their localization near speckles, DNMT1i-specific peaks are located in regions of twofold higher SON signal compared to all CTCF peaks (*Figure 2G*, *Figure 2—figure supplement 1F*). Notably, this trend is strongest for looping DNMT1i-specific CTCF peaks (*Figure 2G*). This enrichment of looping DNMT1i-specific peaks near speckles is further corroborated by publicly available SON TSA-seq data in wild-type cells (*Figure 2—figure supplement 1G*; *Zhang et al., 2021*). Thus, DNMT1i-specific CTCF peaks reside closer to nuclear speckles than non-DNMT1i-specific peaks.

Strikingly, relative to all CTCF peaks, highly looping CTCF peaks are enriched at nuclear speckles (*Figure 2H*, *Figure 2—figure supplement 1H*, ~5-fold enrichment). More broadly, the number of different loops a CTCF peak engages in and their corresponding strength are positively correlated (Pearson R = 0.2) with SON Cut&Tag signal (*Figure 2I*, *Figure 2—figure supplement 1I*). In addition, we observed that CTCF peaks with strong Cut&Tag SON signal preferentially engage in strong shorter-range interactions in published Hi-C interaction data (*Figure 2J*; *Siegenfeld et al., 2022*). Overall, our results reveal that DNMT1i-specific CTCF peaks interact with highly looping partners located near nuclear speckles.

## DNMT1i-induced gene activation and speckle association depend on CTCF

Our data revealed that methylation-dependent CTCF binding and looping are highly correlated with nuclear speckle association. As such, we next sought to study the nature of this relationship through degrading either CTCF or nuclear speckles. We first generated a HaloTAG-CTCF homozygous knock-in HCT116 cell line, which enabled the inducible recruitment of CTCF to the VHL E3 ubiquitin ligase upon addition of the small molecule HaloPROTAC3, causing subsequent proteolytic degradation within 8 hr (*Figure 3A and B*, *Figure 3—figure supplement 1A*). We then performed SON Cut&Tag after pretreating our CTCF-HaloTag knock-in cells with HaloPROTAC3 or vehicle for 8 hr before co-treating with DNMT1i or vehicle for 24 hr (32 hr total for HaloPROTAC3 treatment). This shorter time point was used to capture the direct effects of CTCF degradation and DNA demethylation, as well as to avoid

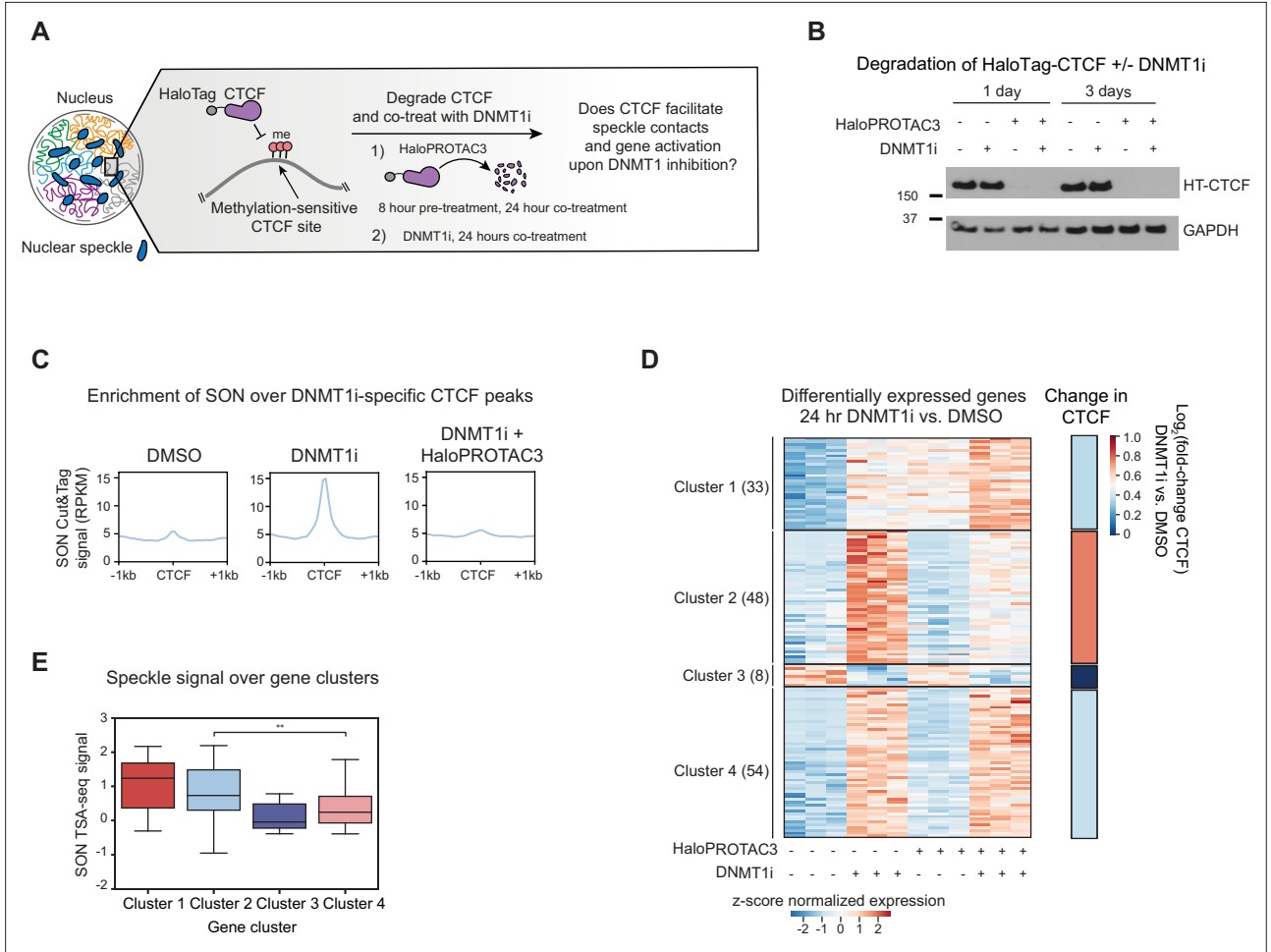

**Figure 3.** DNMT1i-induced gene activation and speckle association depend on CTCF. (**A**) Schematic showing the CTCF degradation experimental workflow. Following pretreatment with HaloPROTAC3 for 8 hr to degrade HaloTag-CTCF, cells are co-treated with HaloPROTAC3 and DNMT1i for 24 hr. DMSO controls without CTCF depletion and/or DNMT1 inhibition were also included. (**B**) Western blot showing the depletion of CTCF in HaloTag-CTCF knock-in cell line in a HaloPROTAC3-dependent and DNMT1i-independent manner (330 nM HaloPROTAC3, 10 μM DNMT1i). (**C**) Aggregate profile of replicate-averaged SON Cut&Tag signal (RPKM) over 1-day DNMT1i-specific CTCF peaks in HaloTag-CTCF HCT116 cells with and without CTCF degradation through HaloPROTAC3. (**D**) Left: Heatmap showing Z-score normalized r-log transformed counts for genes that are differentially expressed in DNMT1i vs. DMSO non-HaloPROTAC3 conditions (p-adj<0.05, |$\log_2$(fold-change)|>0.5) clustered by K-means. The number of genes per cluster is as follows: Cluster 1: 33, Cluster 2: 48, Cluster 3: 6, Cluster 4: 54. Right: Average $\log_2$(fold-change) in CTCF binding in DNMT1i vs. DMSO for all CTCF peaks on genes within the four gene clusters. (**E**) Boxplot showing normalized SON TSA-seq signal (20 kb bins, y-axis, 4D Nucleome 4DNFIBY8G6RZ) in wild-type HCT116 cells over genes within the different clusters identified in D. In (**E**), the interquartile range (IQR) is depicted by the box with the median represented by the center line. Outliers are excluded. P-Values were calculated by a Mann-Whitney test and are annotated as follows: ns: not significant; *: 0.01<p≤0.05; **: 0.001<p≤0.01; ***: 0.0001<p≤0.001; ****: p≤0.0001.

The online version of this article includes the following source data and figure supplement(s) for figure 3:

**Source data 1.** PDF file containing original western blots for *Figure 3B*, indicating the relevant bands and treatments.

**Source data 2.** Original files for western blot analysis displayed in *Figure 3B*.

**Figure supplement 1.** DNMT1i-induced gene activation and speckle association depend on CTCF.

**Figure supplement 1—source data 1.** PDF file containing original western blots for *Figure 3—figure supplement 1A*, indicating the relevant bands and treatments.

**Figure supplement 1—source data 2.** Original files for western blot analysis displayed in *Figure 3—figure supplement 1A*.

toxicity caused by CTCF depletion. We found that DNMT1i-specific CTCF peaks gain ~3-fold SON Cut&Tag signal upon DNMT1i treatment (*Figure 3C*). This gain in SON signal is blocked when CTCF is degraded by HaloPROTAC3 treatment, suggesting that speckle association at these DNMT1i-specific sites is dependent on CTCF (*Figure 3C*). Thus, SON associates with DNMT1i-specific CTCF peaks in a CTCF-dependent manner.

Because DNMT1i-specific peaks fail to engage with nuclear speckles if CTCF is not present, we hypothesized that gene activation induced upon loss of DNA methylation may be dampened in the absence of CTCF. Prior work has suggested that cell-type-specific DNA methylation may protect a 'labile' set of CTCF peaks from binding (*Maurano et al., 2015*) and that methylation-dependent CTCF occupancy on gene bodies has been linked to changes in gene processing, but a global understanding of the functions of methylation-sensitive CTCF peaks on transcription remains lacking (*Shukla et al., 2011*; *Nanavaty et al., 2020*; *López Soto and Lipscombe, 2020*). Thus, we tested whether DNMT1i-specific CTCF peaks promote gene activation upon DNMT1 inhibition.

To assess the role of CTCF in enabling DNMT1i-dependent gene activation, we pretreated our CTCF-HaloTag knock-in cells with HaloPROTAC3 or vehicle for 8 hr and then co-treated the cells with DNMT1i or vehicle for 24 hr before performing RNA-seq (*Figure 3A and D*). Consistent with prior studies (*Nora et al., 2017*), acute CTCF degradation altered the expression of hundreds of genes, with approximately equal numbers up- and downregulated (488 upregulated, 338 downregulated, p-adj<0.05, |log$_2$fold-change|>0.5) (*Figure 3—figure supplement 1B*), with downregulated genes possessing higher baseline levels of CTCF at their promoters (*Figure 3—figure supplement 1C*). As expected, DNMT1i treatment in this cell line for 24 hr primarily resulted in gene upregulation, with 135 genes upregulated and 8 genes downregulated (p-adj<0.05, |log$_2$fold-change|>0.5) (*Figure 3—figure supplement 1B*).

We focused our analysis on genes that were differentially expressed in DNMT1i vs. DMSO treatment in the presence of CTCF (*Figure 3—figure supplement 1B*, left). Using K-means clustering, we identified four gene clusters within this group defined by unique Z-score normalized expression patterns across the conditions (*Figure 3D*). Clusters 1, 2, and 4 comprise genes that increase in expression upon DNMT1 inhibition, whereas Cluster 3 contains the few genes that decrease in expression upon DNMT1 inhibition. Moreover, genes in Cluster 1 are upregulated by CTCF depletion, whereas genes within Clusters 2, 3, and 4 are unaffected by CTCF depletion alone.

We focused on Clusters 2 and 4, which contain lowly expressed genes that are upregulated upon DNMT1 inhibition and unaffected by CTCF depletion in the absence of DNMT1i (*Figure 3D*, *Figure 3—figure supplement 1D*). Significantly, while genes in Cluster 4 are upregulated upon DNMT1 inhibition with or without CTCF present, genes in Cluster 2 depend on CTCF for DNMT1i-mediated upregulation. To determine how CTCF binding might be related to the observed changes in gene expression, we performed CTCF ChIP-seq in our HaloTag-CTCF HCT116 cells following 24 hr of DNMT1i or vehicle treatment. Notably, genes in Cluster 2 preferentially gained CTCF occupancy upon DNMT1 inhibition compared to genes in other clusters (*Figure 3D*, right, *Figure 3—figure supplement 1E*). Thus, these data demonstrate that a subset of genes enriched for methylation-sensitive CTCF binding depends on CTCF for expression upon DNMT1i treatment.

We next assessed the correlation between CTCF-dependent genes and speckle association. Intriguingly, genes in Cluster 2 have nearly threefold higher SON TSA-seq levels than genes in Cluster 4 (*Figure 3E*), while genes in Cluster 1, which are upregulated by CTCF depletion alone, are closest to nuclear speckles. Moreover, genes in Cluster 2 gain a modest (log$_2$fold-change~0.1) level of SON Cut&Tag signal upon DNMT1i treatment (*Figure 3—figure supplement 1F*). Taken together, these experiments show that CTCF is necessary for both gene activation and speckle protein association upon DNMT1 inhibition.

## Acute disruption of nuclear speckles alters RNA abundance without disrupting CTCF

To further explore the observed connection between speckles and CTCF looping, we tested whether nuclear speckles play a causal role in promoting CTCF occupancy and/or looping. To accomplish this, we sought to acutely degrade speckles to evaluate whether speckles actively maintain CTCF loops while mitigating confounding factors associated with mitotic arrest caused by long-term SON depletion (*Ahn et al., 2011*). Consequently, we engineered a K562 knock-in cell line with the core speckle

proteins, SON and SRRM2, both endogenously tagged with FKBP12$^{F36V}$ for use with the dTAG degradation system (*Ilik et al., 2020*; *Nabet et al., 2018*). Upon addition of dTAG-13, FKBP12$^{F36V}$-tagged proteins are recruited to the CRBN-Cul4-Ring E3 ligase and undergo acute proteasomal degradation (*Figure 4A*). Treatment of the K562 FKBP12$^{F36V}$ double knock-in cell line with dTAG-13 resulted in near-complete degradation of both SON and SRRM2 within 6 hr (*Figure 4B*, *Figure 4—figure supplement 1A*).

We next assessed if speckle disruption directly alters RNA abundance by performing RNA-seq after 6 hr of dTAG-13 treatment. Upon speckle degradation, 541 genes were downregulated and 184 genes were upregulated (p-adj<0.05, |log$_2$fold-change|>0.25) (*Figure 4—figure supplement 1B*). Consistent with a role of nuclear speckles in promoting transcription, over 80% of downregulated genes were located within the closest decile to nuclear speckles by TSA-seq (*Figure 4—figure supplement 1C*), and genes with the highest SON Cut&Tag signal exhibited the strongest decrease in RNA abundance (*Figure 4—figure supplement 1D*). These results show that disrupting nuclear speckles has immediate effects on RNA abundance.

To determine whether speckle disruption alters CTCF binding and/or looping, we performed CTCF ChIP-seq and HiChIP in the speckle dTAG cell line after treatment with dTAG-13 or DMSO for 6 hr. In contrast to the immediate RNA effects, CTCF occupancy remained unchanged upon speckle degradation, with only two significantly downregulated peaks (*Figure 4C*, *Figure 4—figure supplement 1E*). CTCF HiChIP also revealed that loop strength was largely unchanged across speckle Cut&Tag and TSA-seq quantiles (*Figure 4D*, *Figure 4—figure supplement 1F*). Even after 12 hr of dTAG treatment, there was only a very subtle decrease in CTCF looping near speckles (*Figure 4—figure supplement 1F*). In addition, changes in gene expression were not associated with changes in loop strength or CTCF binding (*Figure 4—figure supplement 1G and H*). Conversely, depleting CTCF for 24 hr in our HaloTag-CTCF HCT116 knock-in cell line did not noticeably change speckle morphology (*Figure 4—figure supplement 1I*). Altogether, acute disruption of nuclear speckles did not appreciably disrupt CTCF binding or looping, suggesting that maintenance of 3D CTCF loop structure is independent of speckle association. This complements our finding that, conversely, speckle association depends upon DNMT1i-specific CTCF.

## Discussion

DNA methylation accounts for much of the variation in CTCF binding across cell types (*Maurano et al., 2015*), but the functions and features of methylation-sensitive CTCF peaks are incompletely understood. Interestingly, DNA-methylation-sensitive CTCF sites have recently been shown to regulate gene expression during differentiation in mouse embryonic stem cells, supporting the important role of these highly regulated sites in modulating cell state transitions (*Monteagudo-Sánchez et al., 2024a*; *Monteagudo-Sánchez et al., 2024b*). Here, we leveraged a selective DNMT1 inhibitor (*Azevedo Portilho et al., 2021*; *Pappalardi et al., 2021*) to gain new insights into how DNMT1i-specific CTCF peaks interface with chromatin looping and transcription. This new inhibitor is a valuable tool for reactivating CTCF at shorter timescales without the confounding effects caused by the covalent nature and toxicity of earlier hypomethylating agents or the compensatory feedback mechanisms that can result from gene knockout.

We found that DNMT1i-specific CTCF peaks preferentially form loops on gene bodies and interact with promiscuous partner peaks at stripe anchors. Architectural stripes may contribute to transcription by promoting enhancer contacts with gene bodies (*Vian et al., 2018*; *Yoon et al., 2022*; *Barrington et al., 2019*; *Kraft et al., 2019*) and may also represent a mechanism to poise transcription (*Barrington et al., 2019*). Thus, we propose that DNMT1i-specific peaks enable transcriptional activation by interacting with regulatory elements through highly looping CTCF partner peaks (*Figure 4E*). However, DNA demethylation causes pleiotropic effects ranging from changes in gene expression to transcription factor binding (*Azevedo Portilho et al., 2021*; *Yin et al., 2017*), and we cannot rule out that these other features could contribute to the effects we observe.

Strikingly, DNMT1i-specific CTCF peaks and their highly looping partners are located near nuclear speckles, consistent with prior studies, showing that subcompartments near nuclear speckles exhibit increased CTCF binding and looping (*Wang et al., 2021*; *Jabbari et al., 2019*). We first explored this relationship by degrading CTCF and intriguingly observed that DNMT1i-specific CTCF peaks gain SON Cut&Tag signal upon DNMT1 inhibition in a CTCF-dependent manner, indicating a functional tie

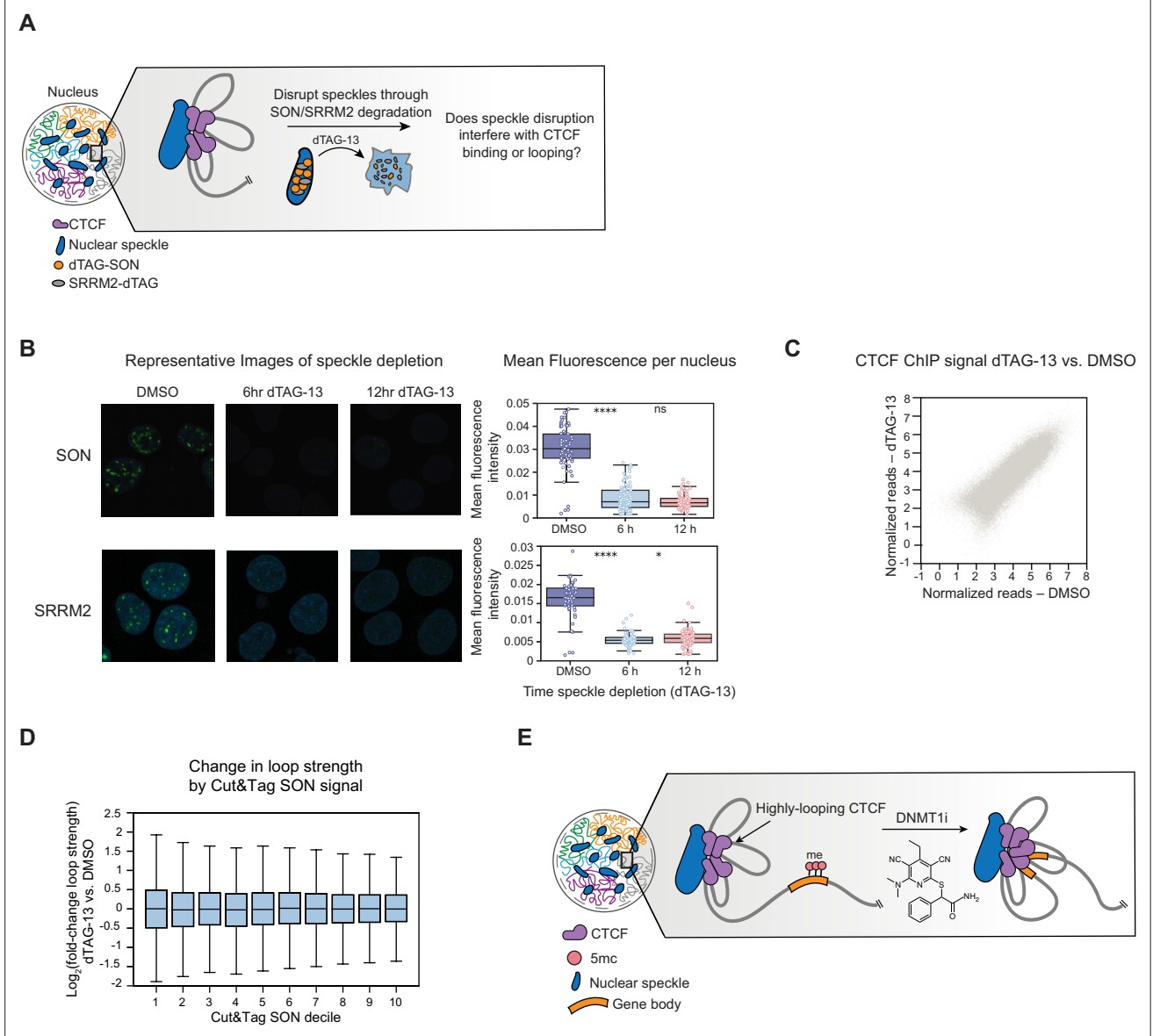

**Figure 4.** Acute disruption of nuclear speckles alters gene expression without disrupting CTCF. (**A**) Schematic illustrating the use of the dTAG system to acutely deplete SON and SRRM2, thus disrupting nuclear speckles. (**B**) Immunofluorescence for SON and SRRM2 in K562 speckle knock-in cells treated with dTAG-13 for 6 or 12 hr to deplete SON and SRRM2. Right: per-nucleus mean fluorescence following immunofluorescence for SON/SRRM2 double dTAG knock-in K562 cells treated with dTAG-13 for 6 or 12 hr or DMSO for 12 hr. (**C**) Replicate-averaged input normalized tags between CTCF ChIP-seq in 6 hr dTAG-13 treatment (x-axis) vs. DMSO (y-axis) for K562 speckle dTAG knock-in cells. (**D**) Boxplot showing $\log_2$(fold-change) in loop strength for CTCF HiChIP loops in speckle knock-in K562 cells treated with dTAG-13 vs. DMSO (y-axis) for 6 hr relative to SON Cut&Tag signal (x-axis). Loops are segregated into equally sized deciles by Cut&Tag signal with 10 representing the decile closest to speckles (RPKM, 20 kb bins). (**E**) Model for methylation-mediated insulation of genes from regulatory elements near nuclear speckle. In (**B and D**), the interquartile range (IQR) is depicted by the box with the median represented by the center line. Outliers are excluded. P-Values were calculated by a Mann-Whitney test and are annotated as follows: ns: not significant; *: $0.01 < p \leq 0.05$; **: $0.001 < p \leq 0.01$; ***: $0.0001 < p \leq 0.001$; ****: $p \leq 0.0001$.

The online version of this article includes the following source data and figure supplement(s) for figure 4:

**Figure supplement 1.** Acute disruption of nuclear speckles alters gene expression without disrupting CTCF.

**Figure supplement 1—source data 1.** PDF file containing original western blots for *Figure 4—figure supplement 1A*, indicating the relevant bands and treatments.

**Figure supplement 1—source data 2.** Original files for western blot analysis displayed in *Figure 4—figure supplement 1A*.

between DNMT1i-specific CTCF and speckles. This is consistent with a recent preprint reporting that CTCF helps establish basal speckle-gene contacts (*Yu et al., 2025*). Additionally, our CTCF degron experiments demonstrate that a large fraction of genes that are differentially expressed upon DNMT1 inhibition depends on CTCF for upregulation. These genes preferentially gain CTCF binding and are located near nuclear speckles. Consequently, CTCF peaks that bind upon DNMT1 inhibition could facilitate the activation of genes near speckles, perhaps by mediating the engagement of genic CTCF sites with speckle-associated stripe anchors and/or splicing factors, although this model requires further study (*Figure 4E*). A role of CTCF in facilitating chromatin loops near speckles is synergistic with proposed models, suggesting that intragenic looping contributes to gene splicing (*Ruiz-Velasco et al., 2017*; *Liao and Regev, 2021*). Thus, DNMT1i-specific CTCF peaks may reorganize genes near nuclear speckles as a means of influencing transcription.

We next studied whether speckles impact CTCF binding or looping by achieving the first acute degradation of nuclear speckles. Previous work knocking down SRRM2 supports a causative role of speckles in modulating chromatin structure but did not specifically study changes in CTCF binding or looping (*Hu et al., 2019*). Although acute disruption of speckles supports their direct involvement in RNA abundance, we observed minimal impacts on CTCF binding or looping. This notion is consistent with recent work, showing that CTCF clustering and looping is not dependent on phase separation or nuclear bodies (*Lee et al., 2022*; *Ulianov et al., 2021*) and that transcriptional disruption does not immediately impact chromatin looping or CTCF binding (*Vian et al., 2018*; *Jiang et al., 2020*; *Luan et al., 2021*). Our findings also suggest that the genome interactions disrupted upon SRRM2 depletion (*Hu et al., 2019*) are not mediated by CTCF or only occur after a longer time. However, it is possible that nuclear speckles contribute to the establishment, but not maintenance, of CTCF loops, which may not be evident in our system due to the short time points employed for speckle degradation. Moreover, other features enriched near speckles, such as open chromatin, high GC content, and active gene expression, could instead contribute to increased CTCF binding and looping near speckles. Overall, our speckle degradation experiments help clarify the directionality of the relationship between speckles and CTCF.

Taken altogether, our findings highlight one possible mechanism through which DNA methylation may modulate genome architecture – by protecting a subset of genic CTCF sites from engaging with promiscuous CTCF peaks located near nuclear speckles. Genes near nuclear speckles may be particularly susceptible to activation through CTCF-dependent mechanisms, and thus might require this layer of protective regulation. Notably, the effects of DNA demethylation cannot be fully uncoupled from gene activation in our experiments, and thus this mechanism will require further study. Furthermore, the GC-rich composition of non-intron gene bodies might increase the likelihood of CTCF motifs in comparison to other genomic regions, increasing the vulnerability of these sites to ectopic CTCF binding. More broadly, our work nominates new functions for the known antagonism between gene body methylation and CTCF occupancy in regulating transcription through remodeling genome architecture.

## Methods
### Cell culture
K562 was obtained from ATCC. HEK293T was obtained from Life Technologies. HCT116 was obtained as a gift from M Shair. Cells were authenticated by Short Tandem Repeat profiling (Genetica) and tested for mycoplasma (Sigma-Aldrich, MP0035). All cell lines were cultured in a humidified 5% $CO_2$ incubator at 37°C with media supplemented with 100 U/mL penicillin and 100 µg/mL streptomycin (Life Technologies, 15070063). K562 was cultured in RPMI-1640 (Life Technologies, 11875119) with 10% fetal bovine serum (FBS) (Peak Serum, PS-FB2). HEK293T was cultured in DMEM (Life Technologies, 1195073) with 10% FBS (Peak Serum, PS-FB2). HCT116 was cultured in McCoy's 5A media (Thermo Fisher Scientific, 16600082) with 10% FBS (Peak Serum, PS-FB2).

### Generating SON, SRRM2, and CTCF knock-in cell lines
To generate an N-terminal Puro-P2A-dtag SON knock-in donor plasmid, SON homology arms of 1 kb were amplified from K562 genomic DNA and cloned into the ph1 backbone Puro-P2A-dTAG, subcloned from pCRIS-PITChv2-Puro-dTAG, a gift from J Bradner and B Nabet (Addgene # 91793).

The SON sgRNA, 5'-CACCGCTGCTCGATGTTGGTCGCCA-3' that was previously used for N-terminal SON knock-in was cloned into PX459 by restriction enzyme digestion (guide design from Allen Institute, https://www.allencell.org/cell-catalog.html). To generate a C-terminal dTAG-P2A-blasticidin SRRM2 knock-in donor plasmid, SRRM2 homology arms of 1 kb were amplified from K562 genomic DNA and cloned into the ph1 backbone dTAG-P2A-BSD, subcloned from pCRIS-PITChv2-dTAG-BSD, a gift from J Bradner and B Nabet (Addgene # 91795). The sgRNA, 5'-CACCGCCATGAGACACCGCT CCTCC-3' that was previously validated for C-terminal SRRM2 knock-in was cloned into PX459 by restriction enzyme digestion (*Ilik et al., 2020*). Notably, this strategy eliminates the last four amino acids of SRRM2, which is necessary due to a sparsity of guides in the region and has been used to study SRRM2 in the context of speckles (*Ilik et al., 2020*). Cells were transfected with both the donor and guide plasmids by nucleofection (Lonza, program T016). Cells were selected 3 days after transfection with puromycin (2 µg/mL) (Thermo Fisher Scientific, A1113803) or blasticidin (8 µg/mL) (Thermo Fisher Scientific, A1113903) for a duration of 5–10 days. Single-cell clones were isolated from the bulk cell line through fluorescence-activated cell sorting. The SON knock-in was generated from wild-type K562 cells, and then the SRRM2 knock-in was introduced on top of the validated SON knock-in clone. To generate the HT-CTCF HCT116 knock-in cell line, the sgRNA 5'-CACCGTGCAGTCGAAGCCAT TGTGG-3' was subcloned into PX459, and HaloTAG-CTCF was subcloned into PDONR223. The guide and donor plasmids were transfected into cells using Fugene HD (Promega). After 7 days, single-cell clones were isolated from the bulk population through fluorescence-activated cell sorting using HaloTag TAMRA labeling to sort potential positive clones.

## Immunoblotting

Pellets were harvested for western blot analysis and washed with PBS (Corning, 21-040-CV). For immunoblots of CTCF, cells were lysed on ice using RIPA buffer (Boston BioProducts) supplemented with fresh HALT Protease Inhibitor (Thermo Fisher Scientific), and the lysates were cleared through centrifugation. The protein concentration of the lysate was determined using the BCA Protein Assay Kit (Thermo Fisher Scientific), and immunoblots for CTCF were performed according to standard procedures with a 6% SureCast acrylamide PAGE gel. For immunoblots of SON and SRRM2 and corresponding loading controls, 1 million cells per condition were lysed on ice for 30 min in 50 µL RIPA buffer (Boston Bioproducts, BP-115) supplemented with 1 mM phenylmethylsulfonyl fluoride (PMSF), 2 mM EDTA, and 1X Halt Protease inhibitors (Thermo Fisher Scientific, 78429). Subsequently, 50 µL of wash buffer was added (50 mM Tris pH 8.0, 2 mM MgCl, 50 mM NaCl, 1 mM) PMSF, 1X Halt Protease inhibitors, 1:500 Benzonase (Millipore Sigma, E1014-25KU), and the samples were rotated at room temperature for 20 min and then clarified by full speed centrifugation at 4°C. The protein concentration of the lysates was determined using the BCA Protein Assay Kit (Thermo Fisher Scientific, 23225). Immunoblotting was performed according to standard procedures with the following modifications: a 3–8% Tris acetate gel (Thermo Fisher Scientific, EA03752) was used, and samples were run for 2 hr 15 min at 80 V at 4°C in 1 X Tris-acetate running buffer (Thermo Fisher Scientific, LA0041) except for the loading control which was run for 90 min. Transfer was conducted in 1X transfer buffer (Research Product International, 501977679) containing 10% methanol and 0.05% sodium dodecyl sulfate (SDS) onto an activated 0.45 µm PVDF membrane (Millipore Sigma, IPVH09120) for 2 hr at 15 V except for loading control which was transferred in 1X transfer buffer with 20% methanol at 12 V for 90 min. The primary antibodies used for immunoblotting are as follows: CTCF (1:2000, Cell Signaling Technology D31H2); SON (1:1000, Abcam ab121759); SRRM2 (1:1000, Thermo Fisher PA5-66827); GAPDH (1:2000, Santa Cruz Biotechnology sc-47724); Vinculin (1:2000, Cell Signaling Technology 13901S).

## Immunofluorescence

Immunofluorescence was performed according to standard protocol. Briefly, K562 speckle knock-in cells treated with 500 nM dTAG-13 or DMSO for the indicated amount of time were cytospun onto glass coverslips for 5 min at 800 × rcf and fixed in 4% PFA for 15 min. HCT116 cells treated with 330 nM HaloPROTAC3 or DMSO were instead grown directly on glass coverslips and fixed in 4% PFA for 15 min. Cells were then washed 3× in PBS, permeabilized with 0.25% Triton X-100 for 10 min, and subsequently washed 3× with PBS. Cells were then blocked in 10% goat serum (MilliporeSigma, G9023) in PBS-T for 1 hr and incubated overnight in 5% goat serum in PBS-T with primary antibody at 4°C (SON [1:300, abcam ab121759] or SRRM2 [1:300, Thermo Fisher PA5-66827]). Following 3×

PBS-T washes, cells were incubated with secondary antibody for 1 hr (1:2000, Invitrogen Alexa Fluor 488 A-11008), washed 3× in PBS-T, and mounted on a coverslip with DAPI (Prolong Diamond Anti-Fade, P36961). Cells were imaged on an LSM180 Confocal microscope. Image quantification was performed using CellProfiler on raw Tiff images to identify nuclei and then quantify average fluorescence signal within each nucleus (*Carpenter et al., 2006*).

## Cell growth assays

K562 or HCT116 cells were plated in triplicate in a 96-well plate. Drug or vehicle (DMSO, Sigma-Aldrich, D8418) was dosed at the specified concentration for 3 days. After 3 days of treatment, cell viability was measured using CellTiter-Glo (Promega, G7572) with the luminescence detector on the SpectraMax i3x (Molecular Devices) plate reader with the SoftMax Pro (version 6.5.1) software, and data were processed using Prism.

## ChIP-seq

CTCF ChIP-seq was performed in duplicate. For the CTCF reactivation experiments, K562 or HCT116 wild-type or HT-CTCF HCT116 knock-in cells were treated with 10 µM GSK3482364 or 0.1% DMSO control for the indicated time. For the speckle depletion experiments, K562 speckle dTAG knock-in cells were treated with 500 nM dTAG-13 or 0.1% DMSO control for 6 or 12 hr. After drug treatment, cells were resuspended and fixed at a density of $10^6$ cells/mL in media containing 10% FBS (Peak Serum, PS-FB2) and 1% formaldehyde (Sigma-Aldrich, F8775) for 10 min with rotation and quenched with 0.2 M glycine and washed in PBS. 5 million cells per condition were lysed on ice for 10 min in 600 µL of ChIP lysis buffer (50 mM Tris-HCl pH 7.5, 1% SDS, 0.25% NaDOC) with HALT protease inhibitors (Thermo Fisher Scientific, 78429). Cells were then diluted to 2 mL with ChIP dilution buffer (50 mM Tris-HCL pH 7.5, 0.01% SDS, 150 mM NaCl, 0.25% Triton X-100, 1X HALT protease inhibitor) and sonicated at 4°C with a Branson Sonicator (45% amplitude, 0.7 s on, 1.3 s off, total time 5 min). The supernatant was clarified by centrifugation at 16,000 × rcf for 10 min at 4°C. The sample was then diluted into 6 mL total with ChIP Dilution Buffer, and the final concentration of Triton was brought to 1%. 50 µL was removed as an input sample and kept at 4°C overnight. Immunoprecipitations were carried out on the remaining sample overnight at 4°C with rotation with 5 µL of antibody (Cell Signaling Technology D31H2 anti-CTCF, 3418S). 50 µL of Protein G Dynabeads (Thermo Fisher Scientific, 10004D) were added to each sample and incubated at 4°C with rotation for 3 hr. Supernatant was discarded on a magnet, and beads were then washed twice with ice-cold RIPA wash buffer (10 mM Tris-HCl pH 8.1, 0.1% SDS, 150 mM NaCl, 0.1 NaDOC, 1% Triton X-100, 1 mM EDTA), once with ice-cold LiCl wash buffer (10 mM Tris-HCl, pH 8.1, 250 mM LiCl, 0.5% Triton X-100, 0.5% NaDOC), and once with ice-cold TE buffer (10 mM Tris-HCl pH 8.0, 50 mM NaCl, 1 mM EDTA). Chromatin was eluted in 100 µL of Elution buffer (10 mM Tris-HCl pH 8.0, 0.1% SDS, 150 mM NaCl, 1 mM EDTA) at 65°C for 1 hr at 1000 rpm. Input samples were brought to 100 µL with Elution buffer and incubated alongside ChIP samples. Supernatant was removed, and 50 µg of RNAse A (Sigma-Aldrich, 10109142001) was added, and the samples were incubated for 30 min at 37°C. Next, 25 µg of Proteinase K (Invitrogen, 25530015) was added, and the samples were incubated at 63°C for 3 hr with 1000 rpm agitation. DNA was subsequently purified by a 2X SPRI (Omega Bio-Tek, M1378-01). Sequencing libraries were then prepared from 5 ng of DNA by performing end-repair with the End-it DNA End-Repair Kit (Lucigen, ER81050), followed by A tailing with NEB Klenow Fragment (3'−5' exo-) (New England Biolabs, M0212), adapter ligation with NEB DNA Quick Ligase (New England Biolabs, M2200), and PCR using NEB Ultra Master Mix (M0544S) and 10X KAPA Library Amp Primer Mix (Kapa Biosystems, KK2623). Samples were sequenced using a NovaSeq SP kit (Illumina) with 50 bp paired-end reads at a sequencing depth of approximately 25 million reads per sample.

## HiChIP

Cells were treated in duplicate. For the DNMT1i experiment, K562 or HCT116 cells were treated with 10 µM GSK3482364 (ChemieTek) or 0.1% DMSO control for 3 days. For the speckle degron experiment, K562 speckle dTAG knock-in cells were treated with 500 nM dTAG-13 (Sigma-Aldrich, SML2601) or 0.1% DMSO control for 6 or 12 hr. CTCF HiChIP was performed on each sample according to a published protocol (*Mumbach et al., 2016*). Specifically, cells were resuspended and fixed at a density of $10^6$ cells/mL in media containing 10% FBS (Peak Serum, PS-FB2) and 1% formaldehyde

(Sigma-Aldrich, F8775) for 10 min with rotation and quenched with 0.2 M glycine and washed in PBS. 10 million cells per condition were resuspended in ice-cold Hi-C Lysis Buffer (10 mM Tris-HCl pH 8.0, 10 mM NaCl, 0.2% NP-40, 1X protease inhibitor) and lysed at 4°C for 30 min. Tubes were centrifuged, and the pellets were washed with 500 µL ice-cold Hi-C lysis buffer. Pellets were then resuspended in 100 µL of 0.5% SDS and incubated at 62°C for 10 min, after which they were quenched with Triton X-100 and incubated at 37°C for 15 min. The samples were then digested with 375 units of DpnII (New England Biolabs, R0543L) for 2 hr at 37°C with shaking at 900 rpm, and then the enzyme was heat-inactivated at 65°C for 20 min. Digested ends were filled in with biotinylated dATP (Life Technologies, 19524-016) for 1 hr at 37°C at 900 rpm, and in situ ligation was performed with T4 DNA ligase (New England Biolabs, M0202L) for 4 hr at room temperature with rotation. Nuclei were pelleted by centrifugation and resuspended in Nuclear Lysis Buffer (50 mM Tris-HCl pH 7.5, 10 mM EDTA, 1% SDS, 1X protease inhibitor) and sheared in Covaris Millitubes (cat # 520135) on a Covaris S220 sonicator with Fill Level 10, Duty Cycle 5, PIP 140, Cycles/Burst 200, for 4 min each. Samples were clarified by centrifugation, and supernatant was split into two tubes and diluted with 2× the volume of ChIP Dilution Buffer (0.01% SDS, 1.1% Triton X-100, 1.2 mM EDTA, 16.7 mM Tris pH 7.5, 167 mM NaCl). 30 µL Pre-washed Protein G Dynabeads (Thermo Fisher Scientific, 10004D) were then added to each tube, and samples were rotated for 1 hr at 4°C. Supernatant was transferred to new tubes, and antibody was added and allowed to bind overnight at 4°C with rotation (15 µL antibody per tube, Cell Signaling Technology D31H2 anti-CTCF, 3418S). 30 µL of pre-washed Protein G Dynabeads were added to each tube and rotated at 4°C for 2 hr. Beads were washed 3× each with Low Salt Wash Buffer (0.1% SDS, 1% Triton X-100, 2 mM EDTA, 20 mM Tris-HCl pH 7.5, 150 mM NaCl), High Salt Wash Buffer (0.1% SDS, 1% Triton X-100, 2 mM EDTA, 20 mM Tris-HCl pH 7.5, 500 mM NaCl), and LiCl Wash Buffer (10 mM Tris pH 7.5, 250 mM LiCl, 1% NP-40, 1% Na-DOC, 1 mM EDTA) at room temperature. Tubes for each sample were recombined, and chromatin was eluted into 100 µL of Elution Buffer (50 mM NaHCO$_3$, 1% SDS) per sample by incubating at room temperature for 10 min with rotation followed by 3 min at 37°C shaking. Elutions were repeated for 200 µL total volume. 10 µL of Proteinase K were added per sample (Invitrogen, 25530015 10 mg/mL), and the samples were incubated at 55°C for 45 min at 900 rpm followed by 67°C for 2 hr at 900 rpm. Samples were purified (Zymogen, D4014) and eluted in 10 µL water and quantified by dsDNA Qubit quantification. Biotinylated DNA was then enriched with 5 µL pre-washed Streptavidin C1 beads per sample (Life Technologies, 65001) in biotin binding buffer (5 mM Tris-HCl pH 7.5, 1 mM EDTA, 1 M NaCl) for 15 min at room temperature with rotation. Beads were washed twice with Tween Wash Buffer (5 mM Tris-HCl pH 7.5, 0.5 mM EDTA, 1 M NaCl, 0.05% Tween 100) at 55°C for 2 min at 900 rpm and were then washed with TD buffer (10 mM Tris-HCl pH 7.5, 5 mM MgCl$_2$, 10% dimethylformamide). Beads were resuspended in TD buffer, and Tn5 (Illumina, 20034198) was added at the suggested dilutions, and tagmentation was allowed to occur at 55°C for 10 min with 900 rpm shaking. Supernatant was removed, and beads were washed in 50 mM EDTA for 30 min at 50°C and twice more for 3 min each. Beads were then washed twice in Tween Wash Buffer at 55°C for 2 min and once in 10 mM Tris. DNA was amplified by PCR with Phusion HF 2X MM and Nextera universal barcodes described previously and was purified by two-sided SPRI purification (Omega Bio-Tek, M1378-01). Samples were sequenced using a NovaSeq SP kit (Illumina) with 100 bp or 50 bp paired-end reads at a sequencing depth of approximately 100 million reads per sample.

## RNA-seq

In triplicate separate cultures, cells were treated with the indicated compounds. For the CTCF degron experiment, HaloTag-CTCF HCT116 knock-in cells were pretreated with 330 nM HaloPROTAC3 (AOBIOUS, AOB36136) or 0.1% DMSO control for 8 hr, and then were treated with 10 µM GSK3482364 or 0.1% DMSO in addition to ±330 nM HaloPROTAC3 for 24 more hours. For the speckle knockdown experiment, K562 speckle dTAG knock-in cells were treated with 500 nM dTAG-13 or 0.1% DMSO control for 6 hr. Total RNA was isolated from 1 million cells per condition using the RNeasy Plus Mini Kit (QIAGEN, 74104). mRNA was isolated using the Poly(A) RNA Selection Kit V1.5 (Lexogen, 157.96), and total RNA seq libraries were prepared using the CORALL Total RNA-Seq Library Prep Kit (Lexogen, 095.24). Samples were sequenced using a NovaSeq SP kit (Illumina) 50 bp paired-end reads at a sequencing depth of approximately 30 million reads per sample for the speckle degron experiments. For the HaloTAG-CTCF degron experiments, samples were instead sequenced to a depth of approximately 15 million reads per sample with 100 bp paired-end reads.

## SON Cut&Tag

For the wild-type cell line experiments, K562 or HCT116 wild-type cells were treated in duplicate with either 10 μM GSK3482364 or 0.1% DMSO control for 3 days. For the CTCF degron experiment, HaloTag-CTCF HCT116 knock-in cells were pretreated with 330 nM HaloPROTAC3 (AOBIOUS, AOB36136) or 0.1% DMSO control for 8 hr, and then were treated with 10 μM GSK3482364 or 0.1% DMSO in addition to ±330 nM HaloPROTAC3 for 24 more hours. Cut&Tag was performed according to published protocol (*Kaya-Okur et al., 2019*). 1 million cells were harvested and washed in PBS. Cells were resuspended in ½ volume ice-cold NE1 buffer (20 mM HEPES-KOH pH 7.9, 10 mM KCl, 0.5 mM spermidine, 0.1% Triton X-100, 20% glycerol with Roche Complete Protease Inhibitor EDTA-free) and lysed on ice for 10 min. Nuclei were resuspended in PBS and fixed with 0.1% formaldehyde (Sigma-Aldrich, F8775) for 2 min at room temperature and quenched with 75 mM glycine. Nuclei were centrifuged, drained, and resuspended in wash buffer (20 mM HEPES-NaOH pH 7.5, 150 mM NaCl, 0.5 mM spermidine, Roche Complete Protease Inhibitor EDTA-free) to ~1 million cells per mL, and 250,000 nuclei per condition (as determined by hematocytometer) were used per condition moving forward. Concanavalin A beads (Cell Signaling Technologies, 93569S) were then activated with binding buffer (20 mM HEPES-KOH pH 7.9, 10 mM KCl, 1 mM $CaCl_2$, 1 mM $MnCl_2$) and 10 μL were added per sample and incubated for 10 min at room temperature with rotation. Liquid was removed on a magnet, and beads were resuspended in 50 μL ice-cold antibody buffer (wash buffer with 2 mM EDTA and 0.1% BSA), and 1 μL of antibody (anti-SON Abcam #ab121759) was added and incubated upright with gentle nutating overnight at 4°C. Beads were resuspended in secondary antibody in wash buffer (1:100 guinea pig anti-rabbit ABIN101961) and tubes were nutated at room temperature for 1 hr. Liquid was removed, and beads were washed three times with wash buffer. pA-Tn5 (purified in-house according to published protocol; *Del Priore et al., 2021*) was mixed 1:50 in Dig-300 buffer (without digitonin, 20 mM HEPES-NaOH pH 7.5, 300 mM NaCl, 0.5 mM spermidine, Roche Complete Protease Inhibitor EDTA-free) and beads were resuspended in 100 μL and nutated at room temperature for 1 hr. Beads were washed three times in Dig-300 buffer. Beads were resuspended in 300 μL Tagmentation Buffer (Dig-300 buffer with 10 mM $MgCl_2$) and incubated at 37°C for 1 hr. 16 mM EDTA, 0.1% SDS, and 50 μg of Proteinase K (Invitrogen, 25530015) were added per sample and incubated at 55°C for 1 hr, and then DNA was purified by PCI-chloroform extraction. DNA was amplified by PCR with NEBNext HiFi 2x PCR Master Mix (NEB M0541S), and the library was purified by 1.3X SPRI purification (Omega Bio-Tek, M1378-01). Samples were sequenced using a NextSeq1000 (Illumina) with 50 bp paired-end reads at a sequencing depth of approximately 10–20 million reads per sample.

## CTCF ChIP-seq data processing and peak calling

Reads were aligned to hg38 using Burrow-Wheeler Aligner (BWA-mem, v 0.7.15) (*Li, 2013*). Sam files were converted to bam files and sorted using Samtools view and sort (v 1.5) (*Li et al., 2009*). Identical ChIP-seq reads were collapsed to avoid PCR duplicates using Picard Tools MarkDuplicates (v 2.18.5) with Java (v 1.8.0) with parameters VALIDATION_STRINGENCY = LENIENTASSUME_SORTED = true REMOVE_DUPLICATES = true. Reads were converted to tdf files for viewing on IGV using Igvtools (v 2.3.95) count with -w 25 (*Robinson et al., 2011*). Tag directories were made from deduplicated bam files with Homer (v. 4.9) makeTagDirectory with Perl (v 5.26.1) (*Heinz et al., 2010*). Peaks were called using Homer (v 4.11) getDifferentialPeaksReplicates, where -t was replicate ChIP tag directories, -i was the corresponding replicate input directories, with parameters -F 10 and -L 4. DNMT1i-specific peaks were called using Homer (v 4.11) getDifferentialPeaksReplicates, where -t was replicate ChIP tag directories for DNMT1i treatment, -b was the replicate ChIP directories for DMSO treatment, and -p was the peaks called in the corresponding DNMT1i treatment condition. These peaks were further restricted to those only called in DNMT1i and not DMSO by using bedtools (v 2.26.0) intersect with the -v and -wa options (*Quinlan and Hall, 2010*). For aggregate analyses, deepTools was used to convert deduplicated bam files to RPKM normalized bigwigs and to compare bigwigs to one another.

## HiChIP data preprocessing and loop calling

HiChIP reads for each sample were aligned to hg38 and processed with HiC Pro (v 3.0.0) using the default settings with a DpnII-digested genome generated with the digest_genome script (*Servant et al., 2015*). For most analyses, HiCPro output.allvalidpairs files for replicate samples were then pooled. Loops were called using Hichipper (v 0.7.7) in call mode with default parameters and the flag

`--input-vi`, and loops for each condition were kept if they had FDR<0.01 and >4 PETs (*Lareau and Aryee, 2018b*). For downstream fold-change and CPM analysis using Diffloop, Hichipper was instead used to call loops from HiCPro outputs without pooling replicates, and the unfiltered loop list was used as input to Diffloop. For all Hichipper analyses, merged peak files containing all peaks from DMSO and DNMT1i for a given cell line were used as the input peak file.

## LIMe-Hi-C analysis

Published LIMe-Hi-C data for three replicates of DMSO and DNMT1 inhibitor treatment conditions in K562 were employed for the analysis (*Siegenfeld et al., 2022*). The same treatment conditions were used as employed in this study. LIMe-Hi-C samples were generated as previously described (*Siegenfeld et al., 2022*) using JuiceMe (v 1.0.0) to generate merged_nodups.txt.gz pairwise contact files. Contacts from all three replicates were subsequently pooled together using the JuiceMe CPU 'mega' script (v 1.0.0). These files were then converted to the cool format at 10 kb resolution using cooler (v 0.8.11). These cool files were normalized using the cooler (v 0.8.11) 'balance' function and were directly utilized for aggregate pileup analysis. To perform pileup analysis, CTCF peaks, identified as described above, were filtered based upon the presence of a CTCF motif using the motif file MA0139.1.tsv.gz downloaded from the JASPAR database and assigned to the motif with the strongest signal using the bioframe (v 0.3.3) 'overlap' function. In addition, CTCF sites that were located within 1000 bp of a previously identified blacklisted region (ENCFF356LFX.bed.gz) were removed using the bioframe 'subtract' function. Pileup analysis was then performed using the cooltools (v 0.5.1) 'pileup' function on peaks deduplicated through the bioframe 'cluster' function with a 'min_dist' threshold set to a resolution of 10 kb to remove duplicate peaks located within the same genomic bin (*Abdennur et al., 2022*). Peaks for which motifs mapped to the – strand were reflected across the diagonal, and the pileup matrix was then plotted in Python (v 3.7) (https://www.python.org/) using standard Matplotlib (v 3.5.2) functions. Highly looping and DNMT1i-specific peaks were defined as specified below. Speckle-associated peaks were defined to be those with an average speckle SON Cut&Tag signal greater than or equal to the 90th percentile genome-wide 20 kb bin average speckle signal.

The stripenn (v 1.1.65.7) (*Yoon et al., 2022*) 'compute' function was used to identify stripes from the merged DNMT1 inhibitor 10 kb cool file with the following parameters: `--norm` KR -m 0.95,0.96,0.97,0.98,0.99. The distances of highly looping peaks and normal-looping peaks to stripe anchors tabulated in the stripenn 'result_filtered.tsv' (p-value <0.1) output were calculated using pybedtools (v 0.8.1) 'closest' function and visualized in Python (v 3.8) (http://www.python.org/) using standard plotting functions. For DNA methylation analysis from LIMe-Hi-C data, CpG methylation fraction bigWigs were quantified over CTCF peaks using deepTools multiBigwigSummary (v 3.5.0).

## CTCF binding scatterplots and aggregate profile plots

To make scatterplots of CTCF normalized tags, RPKM normalized bigwigs were generated for ChIP samples and matched input controls from deduplicated bam files (see above) using deepTools (v 3.4.3) bamCoverage. ChIP files were normalized to their matched input controls using deepTools (v 3.4.3) bigwigCompare with the $\log_2$ option. A merged peak list was generated by combining peak files from DNMT1i and DMSO and collapsing the list with bedtools (v 2.26.0) merge, and normalized ChIP-seq signal per condition was annotated onto these peaks using deepTools multiBigwigSummary. To make aggregate profile plots of CTCF ChIP-seq signal over genes or peak subsets, bigwig files of CTCF ChIP-seq signal were generated for each condition from deduplicated bam files (see above) with deepTools (v 3.4.3) bamCoverage with flag `--normalizeUsing` RPKM. To make aggregate plots over peak subsets, deepTools (v 3.4.3) computeMatrix and plotProfile were used with the following computeMatrix parameters: reference-point `--referencePoint` center -bs 50 a 1000 -b 1000 `--skipZeros`. To make aggregate plots over genes, deepTools (v 3.4.3) computeMatrix and plotProfile were used with the following parameters: `--scale-regions --regionBodyLength` 1000-bs 50 -a 1000 `-b 1000 --skipZeros`. Downstream processing was completed in Python (v 3.7.0) (http://www.python.org/) with Pandas (v 1.0.3), Matplotlib (v 3.2.1), and Seaborn (v 0.11.2).

## Annotating CTCF peaks

CTCF peaks were annotated using Homer (v 4.11) annotatePeaks using the -gtf option with an Ensembl gtf file (v 104) with the standard priority order assigned (*Heinz et al., 2010*). Motifs were

assigned to CTCF peaks using gimmemotifs gimme scan function with -b and -f 0.1 and the JASPAR motif pfm as input. Strand-aware motif methylation heatmaps were made using deepTools. To assess overlap of categories of CTCF peaks with a BED file of predefined subcompartments from SNIPER (*Xiong and Ma, 2019*), bedtools (v 2.26.0) map was used with the – o distinct flag, and data were further processed in Python (v 3.7.0) (http://www.python.org/) and any peaks overlapping multiple subcompartments were discarded. To assess overlap of categories of CTCF peaks with normalized TSA-seq counts, UCSC's function bigWigToBedGraph was first used to convert a bigwig file of normalized SON TSA-seq counts to a bedgraph in 20 kb bins, and then bedtools (v 2.26.0) map was used with – o mean flag to map TSA-seq counts onto CTCF peak list, and the data were further processed in Python (v 3.7.0). The following published files were used from 4D Nucleome for TSA-seq figures: 4DNFIVZSO9RI.bw 4DNFIBY8G6RZ.bw.

## Assigning number of loops (partners) per CTCF peak

CTCF peaks were annotated with the number of different loops (partners) per peak by using pgltools (v 2.2.0) intersect1D to list each loop-peak interception followed by bedtools (v 2.26.0) merge on the list of peaks to sum the number of different loops in which each peak falls within an anchor (*Quinlan and Hall, 2010*; *Greenwald et al., 2017*). To annotate peaks with the number of different loops their maximally looping partner makes, pgltools (v 2.2.0) intersect1D was again used to list each loop-peak interception. Bedtools (v 2.26.0) map was used to annotate the number of different loops the peak in each partner anchor makes from the previous file using the -o max flag. This resulted in one entry per loop that lists the 'original' peak and the number of loops its partner anchor makes. The maximally looping partner per original peak was mapped back onto the original peak list with bedtools (v 2.26.0) map using the -o max flag so that each peak was listed once.

## Calling and characterizing highly looping peaks

Highly looping peaks were called by adapting code used previously to define methylation-rich regions and superenhancers (*Cai et al., 2020*). Specifically, peaks were ordered by rank with respect to the number of different loops they make. The tangent line was drawn where the slope becomes 1 and any peaks above this cutoff were considered highly looping for a given condition. To generate aggregate profile plots of public K562 histone modification bigwig files over CTCF peaks, deepTools (v 3.4.3) computeMatrix was used with the following parameters: reference-point `--referencePoint` center -bs 5 0-a 5000 -b 5000 `--skipZeros`, and results were visualized with deepTools (v 3.4.3) plotProfile. Overlap of peaks with subcompartments and TSA-seq was performed as described above for all CTCF peaks. The following published files were used from the ENCODE consortium for the peak characterization pileups: ENCFF239EBH.bigWig, ENCFF469JMR.bigWig, ENCFF600FDO.bigWig.

## Calculating looping changes across conditions

To calculate fold-change and mean CPM loop strength between conditions, mango Hichipper output files for individual replicates were input into Diffloop (v 1.16.0) and processed with the loopsmake mango function (*Lareau and Aryee, 2018a*). The full loop list was then filtered for loops with FDR<0.01 and with at least 8 PETs across samples for a given cell type, and loops that appeared strongly (>5 PETs) in only one replicate but 0 PETs in the other were removed. The quickAssoc function was then used to get loop strength fold-change and CPM values for each loop. Diffloop output files obtained from the summary() function were then intersected with genomic features (such as DNMT1i-specific peak overlap, subcompartment, Cut&Tag signal) for further analyses using pgltools (v 2.2.0) intersect1D and redundant loops were merged using pgltools (v 2.2.0) merge and files were further processed in Python (v 3.7.0) (http://www.python.org/) with Pandas (v1.0.3), Matplotlib (v 3.2.1), Seaborn (v 0.11.2), and Statannot (v 0.2.3) (*Greenwald et al., 2017*).

## Cut&Tag data processing

Paired-end reads were aligned to hg38, deduplicated, and converted to bam and tdf files as described above for ChIP-seq. To generate aggregate profile plots of SON Cut&Tag signal over CTCF peaks, normalized bigwig files averaging both replicates were generated from deduplicated bam files with deepTools (v 3.4.3) bamCompare with `--operation` mean `--normalizeUsing` RPKM `--scale-FactorsMethod` None. Then, deepTools (v 3.4.3) computeMatrix and plotProfile were used to

plot SON Cut&Tag signal over categories of CTCF peaks with the following computeMatrix parameters: reference-point --referencePoint center -bs 50 -a 1000 -b 1000 –skipZeros. To compare Cut&Tag binned signal to peak and gene sets, replicate-averaged bedgraph files were generated with deepTools (v 3.4.3) bamCompare with --normalizeUsing RPKM --operation mean -of bedgraph --binSize 20000 --scaleFactorsMethod None. Binned Cut&Tag signal was mapped onto genes/peaks with bedtools map (v 2.26.0) with -o mean. To compare loop strength and loop fold-change from Diffloops summary file (described above) to binned Cut&Tag signal, the replicate-averaged Cut&Tag bedgraphs in 20 kb bins were intersected with loop files using pgltools (v 2.2.0) intersect1D with -wa -wb -allA followed by pgltools merge with -o mean (*Greenwald et al., 2017*). For all plots with deciles, equal deciles for each plot were determined in Python (v 3.7.0) with Pandas (v 1.0.3) qcut with duplicates = 'drop' (*Ramírez et al., 2016*). Subsequent processing for all plots was completed in Python (v 3.7.0) (https://www.python.org/) with Pandas (v1.0.3), Matplotlib (v 3.2.1), Seaborn (v 0.11.2), and Statannot (v 0.2.3).

## RNA-seq processing and differential expression analysis

RNA-seq data were analyzed following Lexogen's recommended protocol. First, UMIs were extracted with umi-tools (v 1.1.1) extract (*Smith et al., 2017*). Adapters were then trimmed with cutadapt (v 2.7) (*Martin, 2011*). Trimmed reads were aligned to hg38 using Star aligner (v 2.6.0) with a genome generated using the Ensembl hg38 primary assembly and Ensembl GTF file v 104 with the following settings: --outFilterMultimapNmax 20 --alignSJoverhangMin 8 --alignSJDBoverhangMin 1 --outFilterMismatchNmax 999 --outFilterMismatchNoverLmax 0.6 --alignIntronMin 20 --alignIntronMax 1000000 --alignMatesGapMax 1000000 -- peOverlapNbasesMin 40 --peOverlapMMp 0.8 --outSAMattributes NH HI NM MD (*Dobin et al., 2013*). Reads were then deduplicated with umi-tools dedup (v 1.1.1) with flags multimapping-detection-method=NH and unpaired-reads=discard. For gene expression analysis, HTseq – count (v 0.11.2) was used to count reads in genes (*Anders et al., 2015*). Count files were then processed in DESeq2 (v 1.28) with R (v 4.0.2) using the standard workflow to get fold-change values and using alpha = 0.05 (*Love et al., 2014*; *Zitovsky and Love, 2019*). Hierarchical clustering was performed on the regularized $\log_2$-transformed counts using Euclidean distance as a metric and the complete clustering method. All genes with both an p-adj<0.05, calculated using the Benjamini-Hochberg correction, and |$\log_2$(fold-change)|>0.5 between wild-type DNMT1i and DMSO were included in the clustered heatmap. The regularized $\log_2$-transformed counts for the variable genes were grouped by K-means clustering (K=4), and the Z-scores for the variable genes, ordered according to their cluster identity, are depicted. For differential exon analysis, DEXSeq (v 1.34.1) was used. Files were then imported into R, and fold-change values were computed according to the standard DEXSeq workflow. BioMart was used to map the genomic ranges for each gene onto RNA-seq differential expression output files. Gene expression changes were compared with Cut&Tag signal and CTCF peak overlap using bedtools (v 2.26.0) map with -o mean. Gene Ontology analysis was completed on the online DAVID portal by filtering for goterm_direct and goterm_BP_5 results with FDRj<0.2 (*Sherman et al., 2022*; *Huang et al., 2009*). Further analysis, including volcano plots, was completed in Python (v 3.7.0) (http://www.python.org/) with Pandas (v 1.0.3), Matplotlib (v 3.2.1), Seaborn (v 0.11.2), and Statannot (v 0.2.3).

## Materials availability statement

Plasmids, cell lines, and reagents from this study available upon request.

## Acknowledgements

We acknowledge members of the Liau lab for useful discussions. We acknowledge J Nelson, C Maesner, and Z Niziolek for assistance with cell sorting. We acknowledge the FAS Division of Science Research Computing Group at Harvard University for assistance with cluster computing. We acknowledge S Johnstone for support with the HiChIP protocol. We acknowledge T Lu for helpful discussions about speckle Cut&Tag. We acknowledge H Roh for providing Protein A Tn5. APS and CL were supported by the Herchel Smith Graduate fellowships. NZL was supported by an NSF Graduate Research Fellowship (grant no. DGE1745303). ALW was supported by a Simmons award from Harvard Center for Biological Imaging. This research was supported by startup funds from Harvard University.

## Additional information

### Competing interests

Brian B Liau: Shareholder and member of the scientific advisory board of Light Horse Therapeutics. The other authors declare that no competing interests exist.

### Funding

| Funder | Grant reference number | Author |
|---|---|---|
| Harvard University | | Shelby A Roseman<br>Allison P Siegenfeld<br>Ceejay Lee<br>Nicholas Z Lue<br>Amanda L Waterbury<br>Brian B Liau |
| National Science Foundation | DGE1745303 | Nicholas Z Lue |

The funders had no role in study design, data collection and interpretation, or the decision to submit the work for publication.

### Author contributions

Shelby A Roseman, Allison P Siegenfeld, Conceptualization, Data curation, Formal analysis, Validation, Investigation, Visualization, Methodology, Writing – original draft, Project administration, Writing – review and editing; Ceejay Lee, Nicholas Z Lue, Amanda L Waterbury, Investigation; Brian B Liau, Conceptualization, Resources, Data curation, Supervision, Funding acquisition, Writing – original draft, Project administration, Writing – review and editing

### Author ORCIDs

Shelby A Roseman ![ORCID] https://orcid.org/0000-0003-4056-4030
Allison P Siegenfeld ![ORCID] https://orcid.org/0000-0001-8599-577X
Ceejay Lee ![ORCID] https://orcid.org/0000-0002-4128-9328
Nicholas Z Lue ![ORCID] https://orcid.org/0000-0002-4236-9127
Amanda L Waterbury ![ORCID] https://orcid.org/0000-0002-4473-2866
Brian B Liau ![ORCID] https://orcid.org/0000-0002-2985-462X

Reviewer #2 (Public review): https://doi.org/10.7554/eLife.102930.3.sa1
Author response https://doi.org/10.7554/eLife.102930.3.sa2

---

## Additional files

### Supplementary files

Supplementary file 1. HiChIP looping information.

MDAR checklist

### Data availability

All sequencing data has been deposited on NCBI Gene Expression Omnibus under the super series accession code GSE217760. Custom code is available at https://github.com/liaulab/CTCF-Speckles (copy archived at *Roseman, 2025*).

The following dataset was generated:

| Author(s) | Year | Dataset title | Dataset URL | Database and Identifier |
|---|---|---|---|---|
| Roseman SA, Siegenfeld AP, Lee C, Lue NZ, Waterbury AL, Liau BB | 2024 | DNA methylation insulates exons from CTCF loops with nuclear speckles | https://www.ncbi.nlm.nih.gov/geo/query/acc.cgi?acc=GSE217760 | NCBI Gene Expression Omnibus, GSE217760 |

The following previously published datasets were used:

| Author(s) | Year | Dataset title | Dataset URL | Database and Identifier |
|---|---|---|---|---|
| Gholamalamdari O, van Schaik T, Wang Y, Kumar P, Zhang L, Zhang Y, Gonzalez GAH, Vouzas AE, Zhao PA, Gilbert DM, Ma J, van Steensel B, Belmont AS | 2019 | TSA-seq against SON protein on K562 | https://data.4dnucleome.org/experiments-tsaseq/4DNEXPTYCQQ1/ | 4D Nucleome, 4DNESN3W5126 |
| Gholamalamdari O, van Schaik T, Wang Y, Kumar P, Zhang L, Zhang Y, Gonzalez GAH, Vouzas AE, Zhao PA, Gilbert DM, Ma J, van Steensel B, Belmont AS | 2019 | TSA-seq against SON protein on HCT116 | https://data.4dnucleome.org/experiments-tsaseq/4DNEXCAJWM6W/ | 4D Nucleome, 4DNEXCAJWM6W |
| Bernstein B | 2011 | H3K9ac ChIP-seq on human K562 | https://www.encodeproject.org/experiments/ENCSR000AKV/ | ENCODE, ENCSR000AKV |
| Bernstein B | 2011 | H3K27ac ChIP-seq on human K562 | https://www.encodeproject.org/experiments/ENCSR000AKP/ | ENCODE, ENCSR000AKP |
| Snyder M | 2020 | K562 ATAC-seq | https://www.encodeproject.org/experiments/ENCSR483RKN/ | ENCODE, ENCSR483RKN |
| Siegenfeld AP, Roseman SA, Roh H, Lue NZ, Wagen CC, Zhou E, Johnstone SE, Aryee MJ, Liau BB | 2022 | Polycomb-lamina antagonism partitions heterochromatin at the nuclear periphery | https://www.ncbi.nlm.nih.gov/geo/query/acc.cgi?acc=GSE180230 | NCBI Gene Expression Omnibus, GSE180230 |
| Benoukraf T | 2018 | HCT116_WT_WGBS | https://www.ncbi.nlm.nih.gov/geo/query/acc.cgi?acc=GSM3317488 | NCBI Gene Expression Omnibus, GSM3317488 |
| Myers R | 2016 | HAIB Whole Genome Bisulfite Sequencing K562 | https://www.encodeproject.org/experiments/ENCSR765JPC/ | ENCODE, ENCFF459XNY |

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
