## [Editor Report · eLife Assessment]

This **valuable** study tested the impact of DNA methylation on CTCF binding in two cancer cell lines. Increased CTCF binding sites are enriched in gene bodies, and associate with nuclear speckles, indicating a role in increased transcription. In the revised work, the inferred association with nuclear speckles has been supported with more **solid** data. These results will be of interest to the epigenetics field.

---

## [Referee Report · Reviewer #2 (Public review)]

Summary:

CTCF is one of the most well-characterized regulators of chromatin architecture in mammals. Given that CTCF is an essential protein, understanding how its binding is regulated is a very active area of research. It has been known for decades that CTCF is sensitive to 5-cystosine DNA methylation (5meC) in certain contexts. Moreover, at genomic imprints and in certain oncogenes, 5meC-mediated CTCF antagonism has very important gene regulatory implications. A number of labs (eg, Schubeler and Stamatoyannopoulos) have assessed the impact of DNA methylation on CTCF binding, but it is important to also interrogate the effect on chromatin organization (ie, looping). Here, Roseman and colleagues used a DNMT1 inhibitor in two established human cancer lines (HCT116 [colon] and K562 [leukemia]), and performed CTCF ChIPseq and HiChIP. They showed that "reactivated" CTCF sites-that is, bound in the absence of 5meC-are enriched in gene bodies, participate in many looping events, and intriguingly, appear associated with nuclear speckles. This last aspect suggests that these reactivated loops might play an important role in increased gene transcription. They showed a number of genes that are upregulated in the DNA hypomethylated state actually require CTCF binding, which is an important result.

Strengths:

Overall, I found the paper to be succinctly written and the data presented clearly. The relationship between CTCF binding in gene bodies and association with nuclear speckles is an interesting result. Another strong point of the paper was combining DNMT1 inhibition with CTCF degradation.

Weaknesses:

The most problematic aspect of the original version was the insufficient evidence for the association of "reactivated" CTCF binding sites with nuclear speckles. This has been more diligently assessed in the revised version.

Comments on revisions:

The authors have adequately addressed my points in this revised version.

---

## [Author Response]

The following is the authors’ response to the original reviews

We thank the reviewers for the constructive comments, which have improved the manuscript. In response to these comments, we have made the following major changes to the main text and reviewer response:

(1) Added experimental and computational evidence to support the use of Cut&Tag to determine speckle location.

(2) Performed new Transmission Electron Microscopy (TEM) experiments to visualize interchromatin granule clusters +/- speckle degradation.

(3) Altered the text of the manuscript to remove qualitative statements and clarify effect sizes.

(4) Performed new analyses of published whole genome bisulfite data from LIMe-Hi-C following DNMT1 inhibition to demonstrate that CpG methylation is lost at DNMT1i-specific gained CTCF sites.

(5) Included citations for relevant literature throughout the text.

These revisions in addition to others are described in the point-by-point response below.

**Reviewer #1 (Public review):**
SummaryRoseman et al. use a new inhibitor of the maintenance DNA methyltransferase DNMT1 to probe the role of methylation on binding of the CTCF protein, which is known to be involved chromatin loop formation. As previous reported, and as expected based on our knowledge that CTCF binding is methylation-sensitive, the authors find that loss of methylation leads to additional CTCF binding sites and increased loop formation. By comparing novel loops with the binding of the pre-mRNA splicing factor SON, which localizes to the nuclear speckle compartment, they propose that these reactivated loops localize to near speckles. This behavior is dependent on CTCF whereas degradation of two speckle proteins does not affect CTCF binding or loop formation. The authors propose a model in which DNA methylation controls the association of genome regions with speckles via CTCF-mediated insulation.StrengthsThe strengths of the study are (1) the use of a new, specific DNMT1 inhibitor and (2) the observation that genes whose expression is sensitive to DNMT1 inhibition and dependent on CTCF (cluster 2) show higher association with SON than genes which are sensitive to DNMT1 inhibition but are CTCF insensitive, is in line with the authors' general model.WeaknessesThere are a number of significant weaknesses that as a whole undermine many of the key conclusions, including the overall mechanistic model of a direct regulatory role of DNA methylation on CTCF-mediated speckle association of chromatin loops.

We appreciate the reviewer’s constructive comments and address them point-by-point below.

(1) The authors frequently make quasi-quantitative statements but do not actually provide the quantitative data, which they actually all have in hand. To give a few examples: "reactivated CTCF sites were largely methylated (p. 4/5), "many CTCF binding motifs enriched..." (p.5), "a large subset of reactivated peaks..."(p.5), "increase in strength upon DNMT1 inhibition" (p.5); "a greater total number....." (p.7). These statements are all made based on actual numbers and the authors should mention the numbers in the text to give an impression of the extent of these changes (see below) and to clarify what the qualitative terms like "largely", "many", "large", and "increase" mean. This is an issue throughout the manuscript and not limited to the above examples.Related to this issue, many of the comparisons which the authors interpret to show differences in behavior seem quite minor. For example, visual inspection suggests that the difference in loop strength shown in figure 1E is something like from 0 to 0.1 for K562 cells and a little less for KCT116 cells. What is a positive control here to give a sense of whether these minor changes are relevant. Another example is on p. 7, where the authors claim that CTCF partners of reactivated peaks tend to engage in a "greater number" of looping partners, but inspection of Figure 2A shows a very minor difference from maybe 7 to 7.5 partners. While a Mann-Whitney test may call this difference significant and give a significant P value, likely due to high sample number, it is questionable that this is a biologically relevant difference.

We have amended the text to include actual values, instead of just qualitative statements. We have also moderated our claims in the text to note where effect sizes are more modest.

The following literature examples can serve as positive controls for the effect sizes that we might expect when perturbing CTCF. Our observed effect sizes are largely in line with these expected magnitudes.

https://pmc.ncbi.nlm.nih.gov/articles/PMC8386078/ Fig. 2E

https://www.cell.com/cell-reports/pdf/S2211-1247(23)01674-1.pdf Fig. 3J,K

https://academic.oup.com/nar/article/52/18/10934/7740592 Fig. S5D (CTCF binding only).

(2) The data to support the central claim of localization of reactivated loops to speckles is not overly convincing. The overlap with SON Cut&Tag (figure 2F) is partial at best and although it is better with the publicly available TSA-seq data, the latter is less sensitive than Cut&Tag and more difficult to interpret. It would be helpful to validate these data with FISH experiments to directly demonstrate and measure the association of loops with speckles (see below).

A recent publication we co-authored validated the use of speckle (SON) Cut&Run using FISH (Yu et al, *NSMB* 2025, doi: 10.1038/s41594-024-01465-6). This paper also supports a role of CTCF in positioning DNA near speckles. Unfortunately, the resolution of these FISH probes is in the realm of hundreds of kilobases. This was not an issue for Yu et. al., as they were looking at large-scale effects of CTCF degradation on positioning near speckles. However, FISH does not provide the resolution we need to look at more localized changes over methylation-specific peak sites.

Instead, we use Cut&Tag to look at these high-resolution changes. In Figure 3C, we show that SON localizes to DNMT1i-specific peaks only upon DNMT1 inhibition. We further demonstrate that this interaction is dependent on CTCF. In response to reviewer comments, we have now also performed spike-in normalized Cut&Tag upon acute (6 hr) SON degradation to validate that our signal is also directly dependent on SON and not merely due to a bias toward open chromatin.

**Author response image 1. sa2fig1:** 

TSA-seq has been validated with FISH (Chen et. al., doi: 10.1083/jcb.201807108, Alexander et. Al 10.1016/j.molcel.2021.03.006) Fig 6. We include TSA-seq data where possible in our manuscript to support our claims.

We also note that Fig 2F shows all CTCF peaks and loops, not just methylation-sensitive peaks and loops, to give a sense of the data. We apologize for any confusion and have clarified this in the figure legend.

(3) It is not clear that the authors have indeed disrupted speckles from cells by degrading SON and SRRM2. Speckles contain a large number of proteins and considering their phase separated nature stronger evidence for their complete removal is needed. Note that the data published in ref 58 suffers from the same caveat.

Based upon the reviewers’ feedback, we generated Tranmission electron microscopy (TEM) data to visualize nuclear speckles +/- degradation of SON and SRRM2 (DMSO and dTAG). We were able to detect Interchromatin Granules Clusters (ICGs) that are representative of nuclear speckles in the DMSO condition. However, even at baseline, we observed a large degree of cell-to-cell variability in these structures. In addition, we also observe potential structural changes in the distribution of heterochromatin upon speckle degradation. Consequently, we hesitate to make quantitative conclusions regarding loss of these nuclear bodies. In the interest of transparency, we have included representative raw images from both conditions for the reviewers’ consideration.

We also note that in Ref 58 (Ilik et. Al., https://doi.org/10.7554/eLife.60579), the authors show diffusion of speckle client proteins RBM25, SRRM1, and PNN upon SON and SRRM2 depletion, further supporting speckle dissociation in these conditions.

**Author response image 3. sa2fig3:** 

(4) The authors ascribe a direct regulatory role to DNA methylation in controlling the association of some CTCF-mediated loops to speckles (p. 20). However, an active regulatory role of speckle association has not been demonstrated and the observed data are equally explainable by a more parsimonious model in which DNA methylation regulates gene expression via looping and that the association with speckles is merely an indirect bystander effect of the activated genes because we know that active genes are generally associated with speckles. The proposed mechanism of a regulatory role of DNA methylation in controlling speckle association is not convincingly demonstrated by the data. As a consequence, the title of the paper is also misleading.

While it is difficult to completely rule out indirect effects, we do not believe that the relationship between methylation-sensitive CTCF sites and speckles relies only on gene activity.

We can partially decouple SON Cut&Tag signal from gene activation if we break down Figure 4D to look only at methylation-sensitive CTCF peaks on genes whose expression is unchanged upon DNMT1 inhibition (using thresholds from manuscript, P-adj > 0.05 and/or |log2(fold-change)| < 0.5). This analysis shows that many methylation-sensitive CTCF peaks on genes with unchanged expression still change speckle association upon DNMT1 inhibition. This result refutes the necessity of transcriptional activation to recruit speckles to CTCF.

**Author response image 4. sa2fig4:** 

We note the comparator upregulated gene set here is small (~20 genes with our stringent threshold for methylation-sensitive CTCF after 1 day DNMT1i treatment).

However, we acknowledge that these effects cannot be completely disentangled. We previously included the statement “other features enriched near speckles, such as open chromatin, high GC content, and active gene expression, could instead contribute to increased CTCF binding and looping near speckles” in the discussion. In response to the reviewer’s comment, we have further tempered our statements on page 20/21 and also added a statement noting that DNA demethylation and gene activation cannot be fully disentangled. While we are also open to a title change, we are unsure which part of the title is problematic.

(5) As a minor point, the authors imply on p. 15 that ablation of speckles leads to misregulation of genes by altering transcription. This is not shown as the authors only measure RNA abundance, which may be affected by depletion of constitutive splicing factors, but not transcription. The authors would need to show direct effects on transcription.

We agree, and we have changed this wording to say RNA abundance.

**Reviewer #2 (Public review):**
Summary:CTCF is one of the most well-characterized regulators of chromatin architecture in mammals. Given that CTCF is an essential protein, understanding how its binding is regulated is a very active area of research. It has been known for decades that CTCF is sensitive to 5-cystosine DNA methylation (5meC) in certain contexts. Moreover, at genomic imprints and in certain oncogenes, 5meC-mediated CTCF antagonism has very important gene regulatory implications. A number of labs (eg, Schubeler and Stamatoyannopoulos) have assessed the impact of DNA methylation on CTCF binding, but it is important to also interrogate the effect on chromatin organization (ie, looping). Here, Roseman and colleagues used a DNMT1 inhibitor in two established human cancer lines (HCT116 [colon] and K562 [leukemia]), and performed CTCF ChIPseq and HiChIP. They showed that "reactivated" CTCF sites-that is, bound in the absence of 5meC-are enriched in gene bodies, participate in many looping events, and intriguingly, appear associated with nuclear speckles. This last aspect suggests that these reactivated loops might play an important role in increased gene transcription. They showed a number of genes that are upregulated in the DNA hypomethylated state actually require CTCF binding, which is an important result.Strengths:Overall, I found the paper to be succinctly written and the data presented clearly. The relationship between CTCF binding in gene bodies and association with nuclear speckles is an interesting result. Another strong point of the paper was combining DNMT1 inhibition with CTCF degradation.Weaknesses:The most problematic aspect of this paper in my view is the insufficient evidence for the association of "reactivated" CTCF binding sites with nuclear speckles needs to be more diligently demonstrated (see Major Comment). One unfortunate aspect was that this paper neglected to discuss findings from our recent paper, wherein we also performed CTCF HiChIP in a DNA methylation mutant (Monteagudo-Sanchez et al., 2024 PMID: 39180406). It is true, this is a relatively recent publication, although the BioRxiv version has been available since fall 2023. I do not wish to accuse the authors of actively disregarding our study, but I do insist that they refer to it in a revised version. Moreover, there are a number of differences between the studies such that I find them more complementary rather than overlapping. To wit, the species (mouse vs human), the cell type (pluripotent vs human cancer), the use of a CTCF degron, and the conclusions of the paper (we did not make a link with nuclear speckles). Furthermore, we used a constitutive DNMT knockout which is not viable in most cell types (HCT116 cells being an exception), and in the discussion mentioned the advantage of using degron technology:"With high-resolution techniques, such as HiChIP or Micro-C (119-121), a degron system can be coupled with an assessment of the cis-regulatory interactome (118). Such techniques could be adapted for DNA methylation degrons (eg, DNMT1) in differentiated cell types in order to gauge the impact of 5meC on the 3D genome."The authors here used a DNMT1 inhibitor, which for intents and purposes, is akin to a DNMT1 degron, thus I was happy to see a study employ such a technique. A comparison between the findings from the two studies would strengthen the current manuscript, in addition to being more ethically responsible.

We thank the reviewer for the helpful comments, which we address in the point-by-point response below. We sincerely apologize for this oversight in our references. We have included references to your paper in our revised manuscript. It is exciting to see these complementary results! We now include discussion of this work to contextualize the importance of methylation-sensitive CTCF sites and motivate our study.

**Recommendations for the authors:**

**Reviewer #1 (Recommendations for the authors):**
To address the above points, the authors should:(1) Provide quantitative information in the text on all comparisons and justify that the small differences observed, albeit statistically significant, are biologically relevant. Inclusion of positive controls to give an indication of what types of changes can be expected would be helpful.

We have added quantitative information to the text, as discussed in the response to public comments above. We also provide literature evidence of expected effect sizes in that response.

(2) Provide FISH data to (a) validate the analysis of comparing looping patterns with SON Cut&Tag data as an indicator of physical association of loops with speckles and (b) demonstrate by FISH increased association of some of the CTCF-dependent loops/genes (cluster 2) with speckles upon DNMT1 inhibition.

Please see response to Reviewer 1 comment #2 above. Unfortunately, FISH will not provide the resolution we need for point (a). We have confidence in our use of TSA-seq and Cut&Tag to study SON association with CTCF sites on a genome-wide scale, which would not be possible with individual FISH probes. Specifically, since the submission of our manuscript several other researchers (Yu et al, Nat. Struct. and Mol. Biol. 2025, Gholamalamdari et al eLife 2025) have leveraged CUT&RUN/CUT&TAG and TSA-seq to map speckle associated chromatin and have validated these methods with orthogonal imaging based approaches.

(3) Demonstrate loss of speckles upon SON or SRRM2 by probing for other speckle components and ideally analysis by electron microscopy which should show loss of interchromatin granules.

We have performed TEM in K562 cells +/- SON/SRRM2 degradation. Please see response to Reviewer 1 comment #3. Specifically, interchromatin granule clusters are visible in the TEM images of the DMSO sample (see highlighted example above), however, given the heterogeneity of these structures and potential global alterations in heterochromatin that may be occurring following speckle loss, we refrained from making quantitative conclusions from this data. We instead include the raw images above.

(4) The authors should either perform experiments to clearly show whether loop association is transcription dependent or whether association is merely a consequence of gene activation. Alternatively, they should tone down their model ascribing a direct regulatory role of methylation in control of loop association with speckles and also discuss other models. Unless the model is more clearly demonstrated, the title of the paper should be changed to reflect the uncertainty of the central conclusion.

Please see response to Reviewer 1 comment #4 above.

(5) The authors should either probe directly for the effect of speckle ablation on transcription or change their wording.

We have changed our wording to RNA abundance.

**Reviewer #2 (Recommendations for the authors):**
Major:⁃ There was no DNA methylation analysis after inhibitor treatment. Ideally, genome bisulfite sequencing should be performed to show that the DNMT1i-specific CTCF binding sites are indeed unmethylated. But at the very least, a quantitative method should be employed to show the extent to which 5meC levels decrease in the presence of the DNMT1 inhibitor

Response: We have now included analysis of genome wide bisulfite information from LIMe-Hi-C (bisulfite Hi-C) in K562 following DNMT1i inhibition. Specifically, we leverage the CpG methylation readout and find that DNTM1i-specific CTCF sites are more methylated than non-responsive CTCF peaks at baseline. In addition, these sites show the greatest decrease in CpG methylation upon 3 days of DNMT1 inhibition. We include a figure detailing these analyses in the supplement (Fig S1E). In addition, we have added CpG methylation genome browser tracks to (Fig S1D). In terms of global change, we have found that 3 days of DNMT1 inhibitor treatment leads to a reduction in methylation to about ~1/4 the level at baseline.

I am not convinced that CUT&Tag is the proper technique to assess SON binding. CUT&Tag only works under stringent conditions (high salt), and can be a problematic assay for non-histone proteins, which bind less well to chromatin. In our experience, even strong binders such as CTCF exhibit a depleted binding profile when compared to ChIP seq data. I would need to be strongly convinced that the analysis presented in figures 2F-J and S2 D-I simply do not represent ATAC signal (ie, default Tn5 activity). For example, SON ChIP Seq, CUT&Tag in the SON degron and/or ATAC seq could be performed. What worries me is that increased chromatin accessibility would also be associated with increased looping, so they have generated artifactual results that are consistent with their model.

As the reviewer suggested, we have now performed spike-in normalized SON Cut&Tag with DNMT1 inhibition and 6 hours of SON/SRRM2 degradation in our speckle dTAG knockin cell line. These experiments confirm that the SON Cut&Tag signal we see is SON-dependent. If the signal was truly due to artifactual binding, gained peaks would be open irrespective of speckle binding, however we see a clear speckle dependence as this signal is much lower if SON is degraded.

**Author response image 5. sa2fig5:** 

Moreover, in our original Cut&Tag experiments, we did not enrich detectable DNA without using the SON antibody (see last 4 samples-IgG controls). This further suggests that our signal is SON-dependent.

**Author response image 6. sa2fig6:** 

Finally, we see good agreement between Cut&Tag and TSA-seq (Spearman R=0.82). The agreement is particularly strong in the top quadrant, which is most relevant since this is where the non-zero signal is.

**Author response image 7. sa2fig7:** 

Minor points⁃ Why are HCT116 cells more responsive to treatment than K562 cells? This is something that could be addressed with DNA methylation analysis, for example

K562 is a broadly hypomethylated cell line (Siegenfeld et.al, 2022 https://doi.org/10.1038/s41467-022-31857-5 Fig S2A-C). Thus, there may be less dynamic range to lose methylation compared to HCT116.

Our results are also consistent with previous results comparing DKO HCT116 and aza-treated K562 cells (Maurano 2015, http://dx.doi.org/10.1016/j.celrep.2015.07.024). They state “In K562 cells, 5-aza-CdR treatment resulted in weaker reactivation than in DKO cells…” In addition, cell-type-specific responsiveness to DNA methyltransferase KO depending upon global CpG methylation levels, has also been observed in ES and EpiLC cells (Monteagudo-Sanchez et al., 2024), which we now comment on in the manuscript.

⁃ How many significant CTCF loops in DNMTi, compared to DMSO? It was unclear what the difference in raw totals is.

We now include a supplemental table with the HiChIP loop information. We call similar numbers of raw loops comparing DNMT1i and DMSO, as only a small subset of loops is changing.

⁃ For the architectural stripes, it would be nice to see a representative example in the form of a contact plot. Is that possible to do with the hiChIP data?

As described in our methods, we called architectural stripes using Stripenn (Yoon et al 2022) from LIMe-Hi-C data under DNMT1i conditions (Siegenfeld et al, 2022). Shown below is a representative example of a stripe in the form of a Hi-C contact map.

**Author response image 8. sa2fig8:** 

⁃ Here 4-10x more DNMT1i-specific CTCF binding sites were observed than we saw in our study. What are thresholds? Could the thresholds for DNMT1i-specific peaks be defined more clearly? For what it's worth, we defined our DNMT KO-specific peaks as fold-change {greater than or equal to} 2, adjusted P< 0.05. The scatterplots (1B) indicate a lot of "small" peaks being called "reactivated."

We called DNMT1i-specific peaks using HOMER getDifferentialPeaksReplicates function. We used foldchange >2 and padj <0.05. We further restricted these peaks to those that were not called in the DMSO condition.

⁃ On this note, is "reactivated" the proper term? Reactivated with regards to what? A prior cell state? I think DNMT1i-specific is a safer descriptor.

We chose this term based on prior literature (Maurano 2015 http://dx.doi.org/10.1016/j.celrep.2015.07.024, Spracklin 2023 https://doi.org/10.1038/s41594-022-00892-7) . However, we agree it is not very clear, so we’ve altered the text to say “DNMT1i-specific”. We thank the reviewer for suggesting this improved terminology.

⁃ It appears there is a relatively small enrichment for CTCF peaks (of any class) in intergenic regions. How were intergenic regions defined? For us, it is virtually half of the genome. We did some enrichment of DNMT KO-specific peaks in gene bodies (our Supplemental Figure 1C), but a substantial proportion were still intergenic.

We defined intergenic peaks using HOMER’s annotatepeaks function, with the -gtf option using Ensembl gene annotations (v104). We used the standard annotatepeaks priority order, which is TSS > TTS> CDS Exons > 5’UTR exons >3’ UTR exons > Introns > Intergenic.

Maurano et. al. 2015 (http://dx.doi.org/10.1016/j.celrep.2015.07.024) also found reduced representation of intergenic sites among demethylation-reactivated CTCF sites in their Fig S5A. We note this is not a perfect comparison because their data is displayed as a fraction of all intergenic peaks.

⁃ We also recently published a review on this subject: The impact of DNA methylation on CTCF-mediated 3D genome organization NSMB 2024 (PMID: 38499830) which could be cited if the authors choose.

We have cited this relevant review.